# RLVER: Reinforcement Learning with Verifiable Emotion Rewards for Empathetic Agents

**Peisong Wang**[1][*][‡], **Ruotian Ma**[2][*][†], **Bang Zhang**[2][*], **Xingyu Chen**[3][*][‡], **Zhiwei He**[3], **Kang Luo**[2],
**Qingsong Lv**[2], **Qingxuan Jiang**[2], **Zheng Xie**[2], **Shanyi Wang**[2], **Cixing Li**[2], **Yuan Li**[2],
**Fanghua Ye**[2], **Jian Li**[2], **Yifan Yang**[2], **Jia Li**[1], **Zhaopeng Tu**[2][†], **Xiaolong Li**[2]
[1]The Hong Kong University of Science and Technology (Guangzhou)
[2]Tencent Multimodal Department, Digital Human Center    [3]Shanghai Jiao Tong University
ps.wang@connect.hkust-gz.edu.cn, {ruotianma, zptu}@tencent.com

## Abstract

Large language models (LLMs) excel at logical and algorithmic reasoning, yet their emotional intelligence (EQ) still lags far behind their cognitive prowess. While reinforcement learning from verifiable rewards (RLVR) has advanced in other domains, its application to dialogue—especially for emotional intelligence—remains underexplored. In this work, we introduce RLVER, the first end-to-end reinforcement learning framework that leverages verifiable emotion rewards from simulated users to cultivate higher-order empathetic abilities in LLMs. Within this framework, self-consistent affective simulated users engage in dialogue rollouts and produce deterministic emotion scores during conversations, serving as reward signals to guide the LLM's learning. Fine-tuning publicly available Qwen2.5-7B-Instruct model with PPO boosts its Sentient-Benchmark score from 13.3 to 79.2 while largely preserving mathematical and coding competence. Extensive experiments reveal that: (i) RLVER consistently improves multiple dialogue capabilities; (ii) Thinking and non-thinking models show distinct trends—thinking models excel in empathy and insight, while non-thinking models favor action; (iii) GRPO often yields stable gains, while PPO can push certain capabilities to a higher ceiling; (iv) More challenging environments are not always better—moderate ones can yield stronger outcomes. Our results show that RLVER is a practical route toward emotionally intelligent and broadly capable language agents.

## 1 Introduction

The striking progress of large language models (LLMs) has centered on the rational half of human cognition: deductive reasoning in mathematics (Hendrycks et al., 2021b; Cobbe et al., 2021), program synthesis (Guo et al., 2024; Jain et al., 2024), and algorithmic planning (Yao et al., 2023; Zheng et al., 2024a). Yet authentic human intelligence is grounded in *both* IQ and EQ – logical rigor intertwined with nuanced social and emotional understanding. While today's LLMs demonstrate capacity for both, their proficiency in social and emotional reasoning lags behind their logical acumen, as they still stumble when asked to console a distressed friend or to adapt advice to a user's evolving feelings (Zhang et al., 2025a).

Existing dialogue systems typically enhance emotional intelligence through supervised fine-tuning on annotated counseling corpora (Sun et al., 2021; Liu et al., 2021; Zheng et al., 2022) or rule-based templates (van der Zwaan et al., 2012; Peng et al., 2022). However, these approaches suffer from data scarcity, rigid dialogue structures, and limited generalization. Recent successes in Reinforcement Learning from Verifiable Rewards (RLVR) in mathematics, coding, and search demonstrate that base LLMs can acquire new skills purely through RL signals, without requiring supervised warm-up (Zeng et al., 2025; Guo et al., 2025; Hu et al., 2025; Ma et al., 2025). In the context of enhancing dialogue capabilities, reinforcement learning also offers a compelling alternative: rather than imitating static

---

[*]Equal contribution.
[†]Correspondence to: Ruotian Ma <*ruotianma@tencent.com*> , and Zhaopeng Tu <*zptu@tencent.com*>.
[‡]Work done during internships at Tencent.

ground truth, an agent can directly optimize for long-horizon user satisfaction—provided that a stable interaction environment and consistent reward system are in place. However, the exploration of RLVR for enhancing dialogue capabilities faces several key obstacles:

- the lack of a stable, realistic, and scalable environment for multi-turn conversational rollouts;
- the absence of consistent and verifiable reward designs for general-purpose abilities such as emotional intelligence;

We tackle all three challenges with RLVER, the first end-to-end reinforcement learning framework with *verifiable emotion rewards* (RLVER) for cultivating higher-order empathetic abilities in LLMs. Built upon SAGE (Zhang et al., 2025a)—a framework that constructs self-consistent affective user simulators for realistic and automatic dialogue simulation and evaluation—we establish a stable and scalable environment that enables LLMs to continually simulate dialogue rollouts throughout training. In each conversation, the simulated user updates its emotional state after every agent response, emitting an emotion score in $[0, 1]$ as the reward. Changes in the emotion score are consistent and verifiable; each is deterministically derived through principled reasoning steps grounded in the user's persona, dialogue history, conversational context, and goals. By scaling the simulation environment with a wide range of user behaviors and conversation intents, we alleviate reward hacking arising from homogeneous user preferences.

By fine-tuning a Qwen2.5-7B-Instruct model with Proximal-Policy Optimization (PPO), we show that its Sentient-Benchmark score soars from **13.3** to **79.2**, rivaling frontier models while largely preserving mathematical and coding competence. We also experimented with enforcing explicit "thinking" steps before response generation, in order to compare the behaviors of "thinking" and "non-thinking" models during RL training. Extensive experiments reveal the following key findings: (i) RLVER effectively and reliably improves multiple core dialogue capabilities; (ii) thinking and non-thinking models exhibit distinct developmental patterns under certain settings—thinking models tend to enhance empathy and insight, while non-thinking models focus more on action-oriented capabilities; (iii) compared to PPO, GRPO consistently delivers stable and balanced improvements, whereas PPO can occasionally push the upper bounds of specific capabilities; (iv) when examining user simulators as both environment and reward sources in RL training, we find that more challenging configurations do not necessarily yield better outcomes. On the contrary, moderately demanding but well-aligned setups may better support model growth; (v) RLVER shifts model behavior from solution-centric to genuinely empathic styles in Social-Cognition space. Our findings demonstrate that RL with verifiable emotion rewards is a practical path toward emotionally intelligent and broadly capable language agents.

Our contributions are as follows:

1. **RLVER framework.** We propose Reinforcement Learning with Verifiable Emotion Rewards (RLVER), the first RL paradigm to enhance LLMs' empathetic capabilities using on-the-fly verifiable reward signals from a psychologically grounded, self-consistent user simulator.
2. **Empirical advance.** Applying RLVER to a 7B open-source model elevates its Sentient Benchmark score from 13.3 to 79.2—matching much larger proprietary systems—while preserving performance on mathematics and code-generation benchmarks.
3. **Practical insights.** Through comprehensive experiments, we analyze how training strategies, RL algorithm, and environment design affect empathetic capability development, offering insights into when and how RLVER yields robust improvement or desirable outcomes.
4. **Open resources.** We will release code, checkpoints, prompts, and environment scripts to catalyze future research on emotionally intelligent agents.

## 2 REINFORCEMENT LEARNING WITH VERIFIABLE EMOTION REWARDS

Figure 1 illustrates the RLVER framework, which employs reinforcement learning to enhance an agent's empathetic strategies, enabling it to provide effective support for help-seekers in multi-turn dialogues. The system employs verifiable emotion rewards generated by the user simulator SAGE (Zhang et al., 2025a) to guide the RL training process across diverse empathetic contexts, enabling the agent to develop human-like empathic reasoning and strategic responsiveness.

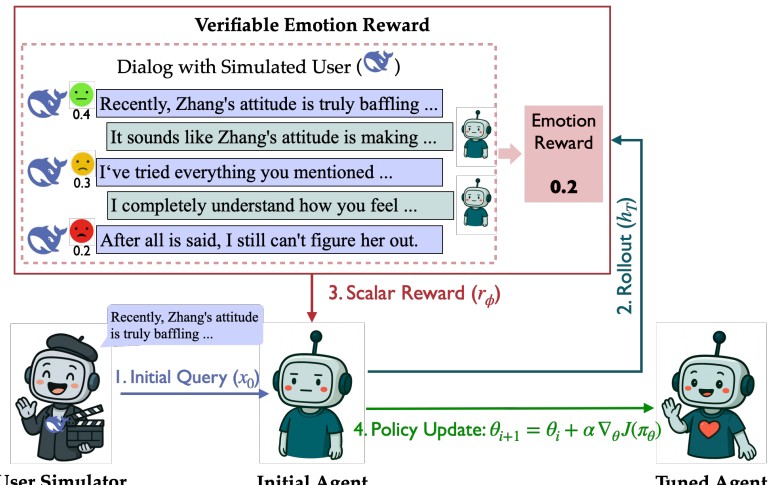

Figure 1: Overview of the RLVER framework. The agent observes an initial query $x_0$ from the user simulator and initiates an interaction rollout, resulting in a dialogue history $h_T$ at termination step $T$. The framework then derives a scalar reward $r_\phi$ based on the simulated user's final emotion score $e_T$. Finally, the normalized reward $r_\phi = e_T/100$ is utilized to update the policy parameters $\theta$ via reinforcement learning.

## 2.1 EMOTION REWARDS FROM SELF-CONSISTENT USER SIMULATION ENGINE

Enhancing general empathetic abilities of LLMs via reinforcement learning requires a dynamic, scalable, and psychologically-grounded environment capable of providing reliable reward signals. Traditional approaches using static datasets or simple LLM-as-a-judge protocols are insufficient, as they fail to capture the user's evolving emotional state throughout a conversation.

To address this gap, our work builds directly upon the **Sentient Agent as a Judge** (SAGE) framework (Zhang et al., 2025a), a sophisticated system designed to automatically evaluate the higher-order social cognition of LLMs. The core of this framework is the **Sentient Agent**, an LLM-powered simulator that mimics human-like emotional responses and inner reasoning. Each agent is instantiated with four key factors: a detailed persona, a dialogue background, an explicit conversation goal, and a hidden intention, ensuring a diverse and realistic range of user simulations.

During an interaction, the Sentient Agent operates in a turn-by-turn loop. After receiving a response from the model being tested, it performs a multi-hop reasoning process to:

- **Simulate Emotional Change** ($f_{\text{emo}}$): The agent assesses how the response made it feel, updating a numerical emotion score and generating interpretable "inner thoughts" that justify the emotional shift.

- **Generate a Coherent Reply** ($f_{\text{reply}}$): Based on its new emotional state, persona, and conversational goals, the agent formulates its own response to continue the dialogue.

The final emotion score in [0, 100] from the Sentient Agent serves as a holistic and quantitative measure of the tested model's empathetic performance. This metric has been shown to correlate strongly with established psychological instruments (e.g., Barrett–Lennard Relationship Inventory) and utterance-level empathy ratings, validating its psychological fidelity.

In our research, we repurpose this evaluation framework as a live training environment. The Sentient Agent acts as the user simulator, and its verifiable, dynamically-generated emotion score provides the crucial reward signal for our reinforcement learning algorithm. Specifically, to alleviate the pervasive challenge of reward hacking in neural reward models during large-scale RL training (Guo et al., 2025), we adopt deterministic emotion scores from the user simulation engine $\mathcal{S}$ as interpretable proxies for simulated user satisfaction, thereby deducing the opacity pitfalls of learned reward functions. Specifically, at each turn $t$ in a conversation, the emotion score $e_t \in [0, 100]$ is updated after each

LLM response $y_t$, reflecting the simulated user's affective state. The final reward is computed as the terminal emotion score normalized by its maximum value:

$$r_\phi(h_T) = \frac{e_T}{100}, \quad \text{where } e_T = \mathcal{S}_{\text{emotion}}(h_T),$$

where $h_T = \{x_0, y_0, \ldots, y_T, x_T\}$ denotes the complete dialogue history at termination ($t = T$) and $\mathcal{S}_{\text{emotion}}(\cdot)$ denotes the function in $\mathcal{S}$ that outputs the final emotion score $e_T$. For intermediate steps, the instantaneous emotion scores $e_t$ track the turn-by-turn evolution of the user's affective state, while the normalized final reward $r_\phi$ exclusively captures the overall conversation quality. This final reward serves as a holistic proxy for user satisfaction at the end of the entire interaction.

## 2.2 HEART-IN-THE-LOOP REINFORCEMENT LEARNING

To enable emotionally intelligent behavior through reinforcement learning, we establish a closed feedback loop whereby the LLM alternates between generating emotionally aware responses and receiving affect-sensitive feedback from the simulation engine. This cycle forms the basis of our Heart-in-the-Loop training paradigm.

Each training step unfolds as a sequence of model-user interactions. At the start of a step $i$, the simulated user engine $\mathcal{S}$ substantiates a batch of Sentient Agents, each initialized by sampling a unique persona, background, emotional tone, and a scenario-driven intention. During each conversation rollout, the agent explores dialogue strategy through live interaction with the substantiated Sentient Agents. Formally, at each time step $t$ in a conversation, the agent observes the current interaction history $h_{t-1}$ and generates a candidate action (response) $y_t \sim \pi_\theta(\cdot \mid h_{t-1})$. The simulator then computes two outputs through principled reasoning:

1. the verifiable emotion score $e_t$, based on the inference of internal emotional state with explicit reasoning from the input context, after receiving $y_t$.

2. a new, contextually coherent user reply $x_t$ based on its updated emotional state, emotional inner thoughts, persona, and conversational goals.

The conversation proceeds until a maximum turn limit $T$ or until the simulator's cumulative emotion score $e_t$ falls below a minimal satisfaction threshold (e.g., $e_t \leq 0$), indicating failed social alignment. The final emotion score $e_T$ serves as the reward function for the reinforcement learning algorithm.

This loop allows the empathetic agent to co-adapt with the simulator's emotional dynamics, progressively learning to map diverse situations, intents, and moods to emotionally satisfying dialogues. By optimizing against a transparent and verifiable reward signal from an emotionally-aware user model, the framework establishes a reproducible and stable setup for training emotionally intelligent LLMs.

**Policy Optimization** For policy optimization, we employ Proximal Policy Optimization (PPO) (Schulman et al., 2017), an on-policy algorithm suited for high-variance environments like language modeling. PPO maximizes a regularized expected reward objective while ensuring stable updates via a clipped surrogate loss. Specifically, the objective function is:

$$L_{\text{PPO}}(\theta) = \hat{\mathbb{E}}_t \left[ \min \left( r_t(\theta) \hat{A}_t, \text{clip}(r_t(\theta), 1 - \epsilon, 1 + \epsilon) \hat{A}_t \right) \right] \tag{1}$$

Here, $r_t(\theta)$ denotes the probability ratio $\frac{\pi_\theta(a_t|s_t)}{\pi_{\theta_{\text{old}}}(a_t|s_t)}$, $\hat{A}_t$ is the estimated advantage at timestep $t$, and $\epsilon$ is a hyperparameter that defines the clipping range.

Benefits of PPO in our setting include safer exploration of diverse social-emotional strategies and smoother convergence when applied alongside the structured thinking scaffold. Additionally, as we aim to optimize the policy for long-term dialogue strategy with outcome-level reward, we also employ Group Relative Policy Optimization (GRPO) (Shao et al., 2024), a promising baseline for learning sequence-level strategies with group-level advantage estimate. This comparison helps assess how learning dynamics respond to different policy gradient estimators in emotionally keyed environments.

While prior zero-RL work has shown that a model can learn from scratch given a well-formed reward function (Zeng et al., 2025), we find that initializing from a modestly aligned checkpoint, pre-trained

with generic conversational data, establishes stronger baselines and accelerates convergence. Notably, this initialization does not require domain-specific supervision—and critically, contains minimal emotional or empathetic signal—ensuring that improvements stem from reward-driven optimization rather than pre-encoded affective knowledge.

# 3 EXPERIMENT

## 3.1 EXPERIMENTAL SETUP

**Base Model**   We adopt Qwen2.5-7B-Instruct (Team, 2024) as our base model, since it is not fine-tuned on domain-specific datasets related to emotional support or empathy. This ensures that any observed improvements in empathetic capability can be attributed to our reinforcement learning process with verifiable emotion-based rewards, rather than prior exposure to affective dialogue data.

**Training Strategy**   During training, we employed two templates to investigate the effect of explicit reasoning: (1) a "think-then-say" template, which elicits the model's reasoning before its reply, and (2) a control template that prompts for a direct reply. The detailed prompts are provided in Appendix A.5.

**Training Environment and Reward**   We adopt the **SAGE** (Zhang et al., 2025a) framework to simulate emotionally responsive users with interpretable affective dynamics and predefined conversational goals. At each turn, the model generates a supportive response, after which the sentient agent replies and updates its internal emotion score $e_t \in [0, 100]$, quantifying its affective state in response to the model's behavior. We scale the final emotion score at the end of the dialogue to the range $[0, 1]$ and use it as the reward for the entire dialogue. Dialogues proceed until the emotional goal is met or a maximum of 8 turns is reached.

We construct a dataset of 500 supportive dialogue scenarios spanning 8 diverse user goals, with topics including emotional struggles, academic stress, interpersonal conflict, and future planning. Unless otherwise specified, we used DeepSeek-V3-1226 (Liu et al., 2024) as the default sentient agent during both training and evaluation. Detailed prompts and additional settings are provided in Appendix A.

**Baselines**   We compare our method against a suite of strong baselines drawn from the top-5 performing models on the SAGE benchmark as of June 9, 2025. These include proprietary state-of-the-art systems Gemini2.5-Pro-0605, GPT-4o-0326, GPT-4.1-0414, Gemini-2.5-Flash-Think-0520, and OpenAI-o3-0416. These models represent the current frontier in instruction-tuned LLMs capable of emotionally sensitive dialogue, and serve as high-performance references for evaluating empathy.

We also include our base model Qwen2.5-Instruct-7B, prior to any further training. This baseline allows us to isolate the contribution of our training strategy, and to establish a controlled comparison against both stronger pretrained models and our own enhanced variants.

**Evaluation Benchmarks**   To evaluate the models' performance in emotionally sensitive dialogue scenarios, we primarily rely on the SAGE benchmark (Zhang et al., 2025a), which focuses on emotional support conversations. To provide a more comprehensive assessment of the model's dialogue capabilities in cross-domain scenarios, we additionally design a "Chit Chat" setting that extends the SAGE framework beyond emotional topics to cover more general, everyday interactions. Furthermore, to examine the potential impact of training on the models' general capabilities, we evaluate its performance on MATH500 (Lightman et al., 2024), LiveCodeBench (Jain et al., 2024), and IFEval (Zhou et al., 2023), which test math reasoning, code generation, and instruction-following abilities, respectively. Further details about adopted benchmarks are provided in Appendix A.1.

## 3.2 MAIN RESULTS

Table 1 presents the results of the proposed RLVER.

**RLVER elevates a lightweight 7B model to near-frontier empathetic performance.**   The base model, Qwen2.5-7B-Instruct, struggles significantly on the Sentient Benchmark, scoring only 13.3

Table 1: Performance of our methods on the Sentient Benchmark. "Success" and "Failure" denote the percentages of dialogues concluding with a final emotion score clipped at 100 and below 10, respectively. We also report results on the out-of-domain chit chat to assess generalization performance.

| Model | | Sentient Benchmark | | | Chit Chat | | |
|---|---|---|---|---|---|---|---|
| RL | Think | Score | Success | Failure | Score | Success | Failure |
| *Top-5 Models in Sentient Leaderboard* | | | | | | | |
| Gemini2.5-Pro-0605 | | 82.4 | 55% | 4% | 83.3 | 77% | 11% |
| GPT-4o-0326 | | 79.9 | 51% | 4% | 80.9 | 74% | 17% |
| GPT-4.1-0414 | | 68.2 | 35% | 13% | 77.1 | 65% | 18% |
| Gemini2.5-Flash-Think-0520 | | 66.1 | 39% | 14% | 64.7 | 53% | 27% |
| OpenAI-o3-0416 | | 62.7 | 32% | 14% | 83.0 | 66% | 9% |
| *Our RLVER-Trained Models* | | | | | | | |
| Qwen2.5-7B-Instruct | | 13.3 | 2% | 76% | 37.8 | 27% | 58% |
| **PPO** | ✗ | 61.7 | 24% | 23% | 53.4 | 39% | 37% |
| | ✔ | 79.2 | 42% | 9% | 62.1 | 52% | 30% |
| **GRPO** | ✗ | 68.3 | 26% | 10% | 49.2 | 34% | 40% |
| | ✔ | 72.0 | 34% | 10% | 53.0 | 45% | 42% |

with a high failure rate (76% of dialogues). In contrast, our RLVER-trained models demonstrate a remarkable improvement. Our best-performing model, trained with PPO and an explicit thinking step ("PPO + Thinking"), achieves a score of 79.2, representing a nearly six-fold increase over the base model. This result not only drastically increases the success rate from 2% to 42% but also brings our 7B model's performance in line with top-tier proprietary models like Gemini2.5-Pro (82.4), while substantially outperforming others such as Gemini2.5-Flash-Think (66.1) and OpenAI-o3 (62.7). This directly validates our primary contribution: the successful application of RL to enhance multi-turn empathetic dialogue capabilities in LLMs.

**"Thinking" models generally exhibit higher empathetic capabilities than "non-thinking" models after training.** Experimental results show that models trained with a thinking scaffold consistently outperform their non-thinking counterparts on both the Sentient Benchmark and Chit-Chat tasks. When trained with PPO, the thinking model achieves a notable improvement from 61.7 to 79.2, surpassing the non-thinking variant. These results suggest that incorporating an explicit reasoning process may facilitate the emergence of higher-order empathetic strategies in LLMs. To further investigate this phenomenon, in §3.3, we present a detailed evaluation of the models' empathetic behavior, demonstrating that eliciting reasoning enhances both the depth of empathy and the ability to identify users' core concerns.

**While both RL algorithms are effective, GRPO tends to offer greater training stability, while PPO provides a higher performance ceiling.** Our results also reveal a nuanced comparison between the PPO and GRPO algorithms. When training both non-thinking and thinking models, GRPO achieves stable improvements, reaching scores of 68.3 and 72.0 respectively. In contrast, PPO yields lower performance in the non-thinking case (61.7), but enables the thinking model to reach a higher performance ceiling (79.2). In §3.3, we further highlight an intriguing observation: PPO and GRPO induce different patterns in the development of model capabilities.

Table 2: Performance of our proposed methods on general tasks.

| Model | | General Capability | | |
|---|---|---|---|---|
| RL | Think | Math500 | LiveCode | IFEval |
| Qwen2.5-7B | | 77.8 | 26.7 | 70.4 |
| **PPO** | ✗ | 76.2 | 33.3 | 72.3 |
| | ✔ | 76.6 | 28.0 | 68.6 |
| **GRPO** | ✗ | 77.4 | 30.0 | 72.1 |
| | ✔ | 75.2 | 29.3 | 69.7 |

**Specialization in empathetic reasoning is achieved with minimal impact on general capabilities.** A critical aspect of fine-tuning is ensuring that specialization in one domain does not lead to

catastrophic forgetting in others. Our evaluation on out-of-domain benchmarks in Table 2 shows that our training successfully avoids this pitfall. While there is a minor decrease in mathematical reasoning performance on Math500 (from 77.8 to 76.6 for our best PPO model), performance on the LiveCodeBench code-generation benchmark is maintained or even improved (from 26.7 to 28.0). Moreover, the model's ability to follow instructions, as measured by IFEval, remains stable (from 70.4 to 68.6). This demonstrates that our framework can cultivate sophisticated emotional intelligence while preserving the model's core general-purpose functionalities, making it a practical and well-rounded solution.

### 3.3 QUALITATIVE ANALYSIS OF TRAINED AGENTS

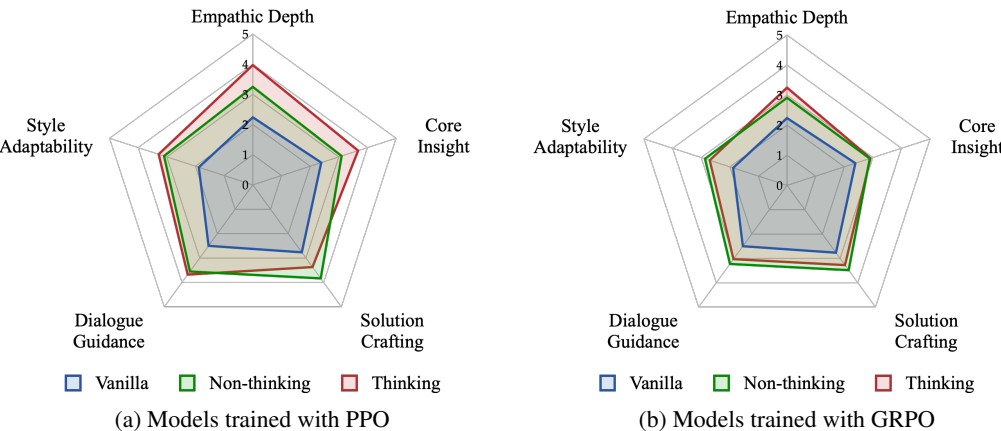

(a) Models trained with PPO

(b) Models trained with GRPO

Figure 2: Qualitative analysis of five core capabilities of the trained models.

To investigate the models' capability improvements after RL training, we formalize a comprehensive evaluation framework encompassing five core competencies in the empathetic dialogue task:

- **Empathic Depth**: Accurately identifying and validating the user's deep emotions with resonant language, moving beyond generic responses.

- **Core Insight**: Synthesizing the user's narrative to uncover the core problem and their unmet emotional needs.

- **Solution Crafting**: Providing actionable, personalized, and empowering suggestions that the user feels capable of implementing.

- **Style Adaptability**: Adjusting its linguistic style and communicative role based on the conversational context and user's needs.

- **Dialogue Guidance**: Proactively steering the conversation from emotional expression toward constructive problem-solving, paced to the user.

We evaluate the five core capabilities using a principled rubric detailed in §B.2. Our evaluation employs an LLM-as-a-Judge framework with five frontier models as judges: DeepSeek-R1-0528, GPT-4o-Latest, Gemini-2.5-Pro, GPT-5-High, and Grok-4. The final score for each capability is the average across these judges. We present detailed results from each judge and a correlation analysis validating the framework's reliability in §B.1.

**RLVER brings consistent improvement across five core capabilities.** As shown in Figure 2, models trained with RLVER—regardless of the specific training strategy—consistently outperform the base model across all five core dimensions. By quantilizing the assessment of these capabilities, we not only gain deeper insight into the behavioral differences induced by different strategies, but also provide an external and objective evaluation—beyond the test set—that supports the effectiveness of our training framework in enhancing key empathetic abilities.

Table 3: Comparison of vanilla and challenging user simulator construction.

| Metric | Vanilla User Simulator | | Challenging User Simulator | |
|---|---|---|---|---|
| | **Non-thinking** | **Thinking** | **Non-thinking** | **Thinking** |
| Sentient Benchmark | 61.7 | 79.2 | 19.8 | 66.4 |
| Sentient Benchmark (challenging) | 47.7 | 59.6 | 25.9 | 44.7 |
| Strategy acceptance rate | 52.4% | | 33.1% | |
| Emotion and need expression level | 78.6% | | 63.6% | |

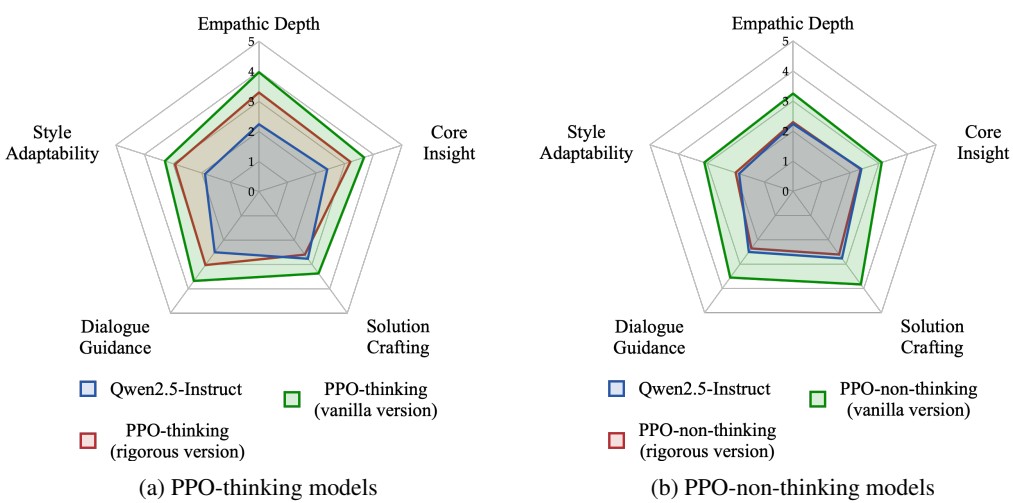

(a) PPO-thinking models        (b) PPO-non-thinking models

Figure 3: Qualitative analysis of training outcomes with vanilla and challenging user simulators.

**Thinking models tend to excel in empathy and insight, while non-thinking models may specialize in action.** With PPO training, we observe a clear divergence in the capability profiles of thinking and non-thinking models. The thinking model exhibits marked improvements in Core Insight (3.673) and Empathic Depth (3.971), demonstrating a strong ability to identify core user needs and to recognize deep emotions through precise, validating responses. In contrast, the non-thinking model shows greater gains in Solution Crafting (3.833), emphasizing actionable, context-aware support through concrete suggestions or behavioral prompts. This pattern suggests that the thinking model benefits from explicit reasoning prior to response generation, enabling it to better infer the user's emotional state and underlying concerns. The non-thinking model, lacking such reasoning, appears to compensate by offering more tangible and personalized solutions to assist the user.

**PPO promotes higher ceilings in specific capabilities, while GRPO supports more balanced and stable development.** A comparison between PPO and GRPO training reveals that GRPO facilitates more balanced and stable improvements across all five capabilities, while PPO tends to amplify specific strengths depending on the training strategy. The thinking model trained with PPO reaches higher performance ceilings in Core Insight (3.673 vs. 2.916 under GRPO) and Empathic Depth (3.971 vs. 3.249), while the non-thinking model trained with PPO achieves a higher ceiling in Solution Crafting (3.833 vs. 3.275). Aligned with the strong performance of the PPO-thinking model in Table 1, these findings suggest that in empathetic dialogue tasks, once a baseline level of competence is achieved across all dimensions (e.g., around 3.0), selectively enhancing high-impact abilities—such as Core Insight and Empathic Depth—may lead to greater practical effectiveness.

## 3.4 IMPACT OF TRAINING ENVIRONMENT AND REWARD

In RLVER, we use self-consistent, scalable user simulators as training environments, with their emotional changes providing rewards. Consequently, training outcomes depend strongly on simulator behavior. After establishing the effectiveness of this setup, we examine how behavioral variations

affect learning by comparing a vanilla simulator with a more challenging variant that is stricter and more reserved. Intuitively, the challenging version demands stronger general capabilities—better identification of unmet emotional needs, deeper empathy, and greater strategic flexibility and dialogue guidance. We characterize the two variants with two metrics: ***Strategy Acceptance Rate*** and ***Emotion and Need Expression Level*** (Table 3; construction details and metric evaluation in Appendix D).

**More challenging environments and reward modeling do not necessarily yield better outcomes**
Table 3 reports PPO results for models trained with vanilla vs. challenging simulators. Models trained with the challenging simulator consistently underperform. On the Sentient Benchmark, the thinking model drops from 79.2 (vanilla) to 66.4; the non-thinking model collapses from 61.7 to 19.8. We also instantiate a challenging version of the benchmark; the gap persists (thinking: 59.6 vs. 44.7; non-thinking: 47.7 vs. 25.9). When simulators serve as both environment and reward, increasing difficulty does not guarantee better learning. Moderately demanding, well-calibrated simulators appear more effective: overly strict/reserved ones restrict feedback during exploration—especially harmful for weaker initial models—while moderate settings provide richer signals that support diverse strategy exploration and broader skill growth.

**Thinking models exhibit greater robustness to environment variations than non-thinking models**
Under the challenging setting, thinking models remain relatively strong (79.2→66.4), whereas non-thinking models suffer severe degradation (61.7→19.8). Figure 3 shows capability development: non-thinking models show little to no improvement, but thinking models still gain in Empathic Depth, Core Insight, and Style Adaptability, indicating resilience under demanding conditions. Notably, the thinking model's gains align with the challenging simulator's behavior: sparse, reserved feedback that rewards deep empathetic reasoning drives targeted improvements in Core Insight (inferring unspoken needs) and Empathic Depth (sensitivity and validation). By contrast, Solution Crafting declines because rollouts seldom reach the suggestion stage, and improvements in Dialogue Guidance and Style Adaptability are limited due to few opportunities to explore dynamic strategies in a restrictive, uncooperative environment.

## 4 RELATED WORK

**Emotional Support Conversation**  Research in Emotional Support Conversation (ESC) has advanced through the development of specialized datasets and models. Early datasets, often derived from psychotherapy transcripts and online forums, were limited to single-turn interactions and narrow contexts (Medeiros & Bosse, 2018; Sharma et al., 2020). The ESConv dataset introduced multi-turn, strategic dialogues (Liu et al., 2021), while later work like AUGESC leveraged LLMs to augment data and reduce annotation costs (Zheng et al., 2022). Modeling approaches have evolved from rigid, rule-based systems (van der Zwaan et al., 2012) to data-driven architectures incorporating graph networks and commonsense reasoning (Peng et al., 2022; Tu et al., 2022). More recently, the field has focused on fine-tuning Large Language Models (LLMs) via supervised fine-tuning Liu et al. (2023), dialogue expansion (Chen et al., 2023; Qiu et al., 2023), knowledge distillation (Zheng et al., 2024b), and strategic planning with Monte Carlo Tree Search (Zhao et al., 2025b).

These advancements, however, have largely relied on supervised learning. To our knowledge, no prior work has applied reinforcement learning (RL) to enhance empathic reasoning in LLMs, nor has any study systematically analyzed the trade-off between logical coherence and emotional sensitivity in generated support. Our work addresses these critical gaps.

**"Zero RL" Training**  The "Zero RL" paradigm, popularized by DeepSeek-R1 (Guo et al., 2025), involves applying reinforcement learning directly to a pretrained base LLM without any intermediate supervised fine-tuning. This approach has proven effective across numerous domains, including mathematics (Zeng et al., 2025; Hu et al., 2025; He et al., 2025; Zhang et al., 2025c; Liu et al., 2025), search (Jin et al., 2025; Song et al., 2025), general-purpose reasoning (Cheng et al., 2025b; Huan et al., 2025), and other specialized fields (Su et al., 2025). Recent research has focused on improving data efficiency (Wang et al., 2025b; Zhao et al., 2025a), eliminating the need for external rewards (Zhao et al., 2025c; Zhang et al., 2025b; Agarwal et al., 2025; Yu et al., 2025), investigating emergent reasoning abilities (Yue et al., 2025; Wen et al., 2025), and analyzing the underlying training mechanisms (Cui et al., 2025; Wang et al., 2025a; Zhu et al., 2025; Cheng et al., 2025a). Despite

these successes, the application of Zero RL to conversational systems remains scarce. In this work, we bridge this gap by introducing RLVER, the first RL framework with verifiable emotion rewards for empathetic dialogue. RLVER endows an LLM with empathetic skills through deterministic, transparent reward signals generated on-the-fly by a psychologically grounded user simulator.

## 5 CONCLUSION

In this study, we demonstrate that emotionally intelligent behaviors can be effectively and reliably acquired through RLVER training, even with a medium-scale LLM and without costly human annotation. Our success hinges on two key components: (i) a self-consistent user simulator (Zhang et al., 2025a) that generates verifiable emotion rewards, and (ii) principled, well-calibrated choices in training strategies, RL algorithms, and environment and reward design. The resulting agent matches frontier-scale proprietary models on the Sentient Benchmark, while preserving strong general reasoning abilities. Beyond empathy, RLVER suggests a broader recipe for aligning language agents with complex, human-centered objectives whenever verifiable reward proxies are available. Future work includes richer multi-party simulations, adaptive persona switching, and integrating multimodal affect to realize truly holistic social intelligence.

## ETHICS STATEMENT

Our work adheres to the ICLR Code of Ethics. The primary ethical consideration of this research is the development of AI agents for empathetic dialogue and emotional support. We stress that our framework, RLVER, is a foundational research system intended to explore and enhance the emotional intelligence of language models, and it is not designed or evaluated as a substitute for professional medical or psychological advice and care. Our training and evaluation processes exclusively use the SAGE framework, which relies on simulated users rather than direct interaction with real individuals, thus avoiding potential psychological distress. All data and models used are from publicly available sources, and our work does not involve sensitive personal data.

## REPRODUCIBILITY STATEMENT

We are committed to ensuring the reproducibility of our research. To facilitate this, we will publicly release our code, final model checkpoints, training prompts, and environment scripts (available at: `https://github.com/Tencent/DigitalHuman/tree/main/RLVER`). Our methodology relies on publicly available models, specifically Qwen2.5-7B-Instruct as the base model and DeepSeek-V3 for the user simulator, as detailed in §3.1. In Appendix §A, we provide comprehensive details necessary for replication, including the specific training hyperparameters (Appendix §A.3), software environment with library versions (Appendix §A.4), and the exact "think-then-say" and control prompt templates used during training (Appendix §A.5). We provide detailed description of the training and evaluation scenarios (Appendix §A.2) and the benchmarks used (Appendix §A.1). For our qualitative analysis, we transparently describe our LLM-as-a-Judge framework and provide the complete evaluation rubric for all five core competencies in Appendix §B, enabling others to replicate our evaluation process.

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

# A    DETAILED EXPERIMENT SETTINGS

## A.1    BENCHMARKS

**SAGE** (Zhang et al., 2025a) is a sophisticated system designed to automatically evaluate the higher-order social cognition of LLMs. The core of this framework is the Sentient Agent, an LLM-powered simulator that mimics human-like emotional responses and inner reasoning. Each agent is instantiated with four key factors: a detailed persona, a dialogue background, an explicit conversation goal, and a hidden intention, ensuring a diverse and realistic range of user simulations. Since the talking strategy of LLMs significantly influence the their performance, we set the prompt of top-5 LLMs in the benchmark as concise as possible to avoid introducing human interference. Therefore, the prompt template used for target LLMs is shown as follows:

```
You are an intelligent conversational partner, skilled at conversing with users in a way
that is emotionally intelligent, making them feel comfortable, happy, or providing the
help they need.
```

**Emotional Support Scenario** is the major scenario introduced in **SAGE**. Agents in this scenario aim to seek support through social interactions including seeking advice, emotional comfort, and other forms of support, rather than through professional counseling. Agents are given various types of task-related hidden intentions covering both emotional intentions and rational intentions are included. Additionally, each conversation background is Carefully designed with incorporating task-related factors, such as the cause of the event, the course of events, the conflicts in the event, and other relevant details.

**Chit Chat Scenario** is an extension of **SAGE**. In contrast to the Emotional Support Scenario, its primary focus is on simulating daily chatting dialogue. This allows for the evaluation of the model's conversational skills, including its ability to be engaging and coherent. Agents are give task-related hidden intentions such as "interest-driven chatting" and "passively waiting chatting", which presents a significant test of the model's ability to adapt its strategy.

**MATH500** (Lightman et al., 2024) offers a streamlined slice of the broader MATH Hendrycks et al. (2021a) dataset, comprising 500 test problems selected through uniform sampling. Despite its smaller scope, it maintains a distribution of topics and difficulty levels that mirrors the larger MATH corpus.

**LiveCodeBench** (Jain et al., 2024) provides holistic and contamination-free evaluation of coding capabilities of LLMs. Particularly, LiveCodeBench continuously collects new problems over time from contests across three competition platforms – LeetCode, AtCoder, and CodeForces. Next, LiveCodeBench also focuses on a broader range of code-related capabilities, such as self-repair, code execution, and test output prediction, beyond just code generation. We use version "release_v6" in this work.

**IFEval** (Zhou et al., 2023) is a clear and reproducible evaluation benchmark that centers on a set of "verifiable instructions", such as "write more than 400 words" and "mention the keyword AI at least three times." A total of 25 types of these verifiable instructions were identified, and approximately 500 prompts were created, each containing one or more verifiable instructions. We report the "strict-prompt" results in this work.

## A.2    EMOTIONAL SUPPORT SCENARIO SETTING

We construct 500 supportive dialogue scenarios for training and 100 for testing. Both the training and test sets span 8 diverse topics to comprehensively simulate individuals with varying emotional needs. Detailed statistics for each topic are presented in Table 4.

## A.3    HYPERPARAMETERS SETTING

We use a batch size of 32 and set the learning rate to $1 \times 10^{-6}$. A warm-up phase of 50 steps is applied. The number of dialogue turns is fixed at 8. The sampling temperature of trained Qwen2.5-7B-Instruct is set to 1 to encourage exploration. For PPO, the rollout sampling number is set to 1, while it is set to 4 for GRPO. The DeepSeek-V3-1226 API is used as the base model for the user simulator.

| Topic | #Training | #Testing |
|---|---|---|
| You believe you bear no responsibility or fault in the situation, and you want the other person to agree that you are not at fault. | 66 | 11 |
| You hope the other person will guide you to engage in self-reflection regarding the incident and help you achieve personal growth. | 66 | 13 |
| You hope the other person will critically analyze the underlying problems in the incident. | 66 | 12 |
| You hope the other person will deeply empathize with your feelings, rather than simply offering comfort. | 64 | 13 |
| You want the other person to attentively listen to your emotional outpouring. | 63 | 12 |
| You want to analyze the reasons behind the actions of other individuals involved in the incident. | 60 | 11 |
| You hope to receive advice that can genuinely help you overcome your current difficulties. | 58 | 15 |
| You hope the other person will sincerely praise your specific actions in the situation. | 57 | 13 |

Table 4: Details of supportive dialogue topics.

## A.4 EXPERIMENTAL ENVIRONMENT

All experiments are implemented using PyTorch 2.5.1 and Ray 2.24.1. Our training code is built upon verl (Sheng et al., 2025).For inference, we use vLLM-0.6.6 (Kwon et al., 2023). We utilize transformers version 4.48.3.

## A.5 THINK-THEN-SAY FOR ENHANCED EMOTIONAL REASONING

To investigate the impact of explicit reasoning on the empathetic strategies, we conduct an ablative analysis using two distinct training templates. These templates structure the agent's generation process, allowing us to isolate the effect of a mandated "think-then-say" cognitive scaffold.

**Think-Then-Say** One of the key innovations in RLVER is the use of a structured "think-then-say" prompting template. This involves including an explicit `<think>...</think>` block before every model utterance during training, compelling the model to outline its reasoning process before delivering a response.

This template, shown below, enforces an explicit chain-of-thought reasoning step. The agent is instructed to first generate its internal monologue or strategic plan within a pair of `<think>` and `</think>` tags before producing the final, user-facing reply. This structure is designed to encourage the model to access and refine higher-order empathetic skills, such as considering the user's emotional state, anticipating the impact of its words, and formulating a multi-step conversational plan. By externalizing its reasoning process, the model's policy space is regularized, potentially leading to more stable learning and more sophisticated final behaviors.

---

**Training Template of Think-Then-Say**

You are chatting with your friend. You are good at making your friend feel better through emotionally intelligent replies. Before each reply, you always think about the way and content of your response; after deciding on a reply strategy, you then output your reply.

Your goal in replying is to improve your friend's mood or to make your relationship with them closer.

In your thinking process, you need to consider emotionally intelligent reply strategies, which can include the logic and language style of your response. Your thinking part must be enclosed within `<think>` tags.

When replying, you should keep the conversation warm and natural, with a casual, everyday feel.

Your reply format: `<think>` {Your thoughts} `</think>` {Your reply}

---

We also employ a **format reward** that enforces the model to put its thinking process between `<think>` and `</think>` tags. Outputs violating this syntactic specification are penalized with zero reward, ensuring strict adherence to the prescribed reasoning structure.

**Template Without Think.** This template serves as our control condition. As shown below, it omits the requirement for an explicit thinking step and prompts the agent to generate a direct reply. This configuration mirrors standard conversational fine-tuning setups. By comparing the performance of models trained with and without the thinking scaffold, we can empirically measure the contribution of the explicit reasoning step to overall empathetic proficiency, learning efficiency, and strategic depth.

---

**Training Template Without Thinking**

You are chatting with your friend. You are good at making your friend feel better through emotionally intelligent replies.

Your goal in replying is to improve your friend's mood or to make your relationship with them closer.

When replying, you should keep the conversation warm and natural, with a casual, everyday feel. Natural and friendly replies usually:
1. Are brief, casual, and natural, using everyday words or phrases; grammar can be flexible.
2. Use interjections and colloquial expressions flexibly.

---

During training, the think-then-say scaffold acts as an internal planning regularizer, guiding the model to first consider its intentions, linguistic tone, and potential emotional impact before forming a conversational reply. We observe that agents trained with this prompting format converge faster, exhibit greater linguistic diversity, and more reliably explore high-empathy strategies.

By contrast, models trained without structured thinking tend to converge to safe, generic replies (e.g., "I'm here for you" or "You're not alone"), which—while emotionally neutral—fail to exhibit situation-specific empathy. Including the reasoning component enables model behaviors to grow beyond templated reassurance and toward goal-sensitive emotional alignment.

## B    EVALUATION OF FIVE CORE MODEL CAPABILITIES

### B.1    DETAILED RESULTS AND VALIDITY VERIFICATION

Our evaluation of core capabilities uses an LLM-as-a-Judge approach with 5 frontier models. The scores for each capability, presented in Section 3.3, were averaged across these judges. In Table 5 , we present the detailed results from each judge individually.

To validate the reliability of our core capability evaluation framework, we calculated the average pairwise correlation between judges on a model-wise basis for each competency. As shown in the final row of Table 5, the results confirm a high degree of inter-judge reliability, strengthening the credibility of the evaluation framework. Specifically, the results show that LLM judges achieve a strong consensus when evaluating key capabilities in emotional support conversations, such as Empathic Depth (correlation of 0.981) and Core Insight (0.943). While the agreement remains substantial across most dimensions, the correlation for Solution Crafting is relatively lower (0.671). We found that different judges hold varying standards for what constitutes "appropriate and personalized action support", making this dimension more subjective.

### B.2    EVALUATION CRITERIA FOR CORE MODEL CAPABILITIES

### B.2.1    EMPATHY DEPTH

Measures the model's ability to go beyond templated responses like "I'm sorry to hear that" to genuinely identify and understand the user's complex, deep-seated emotions, and to **accurately validate** these emotions through **precise, warm, and powerful language**. This reflects the model's emotional granularity and its **ability to construct empathetic language**.

**1-5 Point Evaluation Scale**

- **1 Point (Templated Response)**: Uses extremely generic, context-irrelevant sympathy templates (e.g., "I'm sorry", "I understand"), appearing perfunctory and mechanical.

Table 5: Detailed Results of Core Capability Evaluation.

| Model | Competencies | | | | |
| --- | --- | --- | --- | --- | --- |
| | Empathic Depth | Core Insight | Solution Crafting | Dialogue Guidance | Style Adaptability |
| *Judge Model: DeepSeek-R1-0528* | | | | | |
| Qwen2.5-7B-Instruct | 2.333 | 2.537 | 2.898 | 2.594 | 1.898 |
| PPO-thinking | 3.727 | 3.455 | 3.187 | 3.526 | 3.033 |
| PPO-non-thinking | 3.033 | 3.131 | 3.753 | 3.649 | 3.005 |
| GRPO-thinking | 3.108 | 3.042 | 3.552 | 3.451 | 2.854 |
| GRPO-non-thinking | 2.892 | 2.957 | 3.577 | 3.448 | 2.806 |
| PPO-thinking (challenging) | 3.152 | 3.193 | 2.312 | 2.862 | 2.626 |
| PPO-non-thinking (challenging) | 2.500 | 2.639 | 2.820 | 2.617 | 2.276 |
| *Judge Model: GPT-4o-Latest* | | | | | |
| Qwen2.5-7B-Instruct | 2.093 | 2.198 | 2.384 | 2.163 | 1.698 |
| PPO-thinking | 3.820 | 3.444 | 3.159 | 3.483 | 3.400 |
| PPO-non-thinking | 3.022 | 2.817 | 3.258 | 2.935 | 2.720 |
| GRPO-thinking | 3.086 | 2.882 | 2.871 | 2.804 | 2.720 |
| GRPO-non-thinking | 2.609 | 2.739 | 2.891 | 2.634 | 2.462 |
| PPO-thinking (challenging) | 3.000 | 2.925 | 1.925 | 2.403 | 2.358 |
| PPO-non-thinking (challenging) | 2.189 | 2.226 | 2.170 | 2.000 | 1.811 |
| *Judge Model: Gemini-2.5-Pro* | | | | | |
| Qwen2.5-7B-Instruct | 2.169 | 2.349 | 2.641 | 2.413 | 1.506 |
| PPO-thinking | 4.258 | 3.917 | 4.033 | 4.169 | 3.312 |
| PPO-non-thinking | 3.542 | 3.084 | 4.244 | 3.641 | 3.073 |
| GRPO-thinking | 3.458 | 2.958 | 3.420 | 2.795 | 2.135 |
| GRPO-non-thinking | 3.053 | 2.947 | 3.732 | 3.205 | 2.596 |
| PPO-thinking (challenging) | 3.731 | 3.394 | 3.746 | 3.807 | 3.438 |
| PPO-non-thinking (challenging) | 2.311 | 2.197 | 2.411 | 2.397 | 1.492 |
| *Judge Model: GPT-5-High* | | | | | |
| Qwen2.5-7B-Instruct | 1.989 | 2.122 | 2.511 | 2.111 | 1.589 |
| PPO-thinking | 3.474 | 2.876 | 2.031 | 2.577 | 2.320 |
| PPO-non-thinking | 2.938 | 2.763 | 3.412 | 3.216 | 2.732 |
| GRPO-thinking | 2.844 | 2.458 | 2.469 | 2.260 | 2.042 |
| GRPO-non-thinking | 2.732 | 2.598 | 3.062 | 2.918 | 2.680 |
| PPO-thinking (challenging) | 2.899 | 2.913 | 1.928 | 2.333 | 2.580 |
| PPO-non-thinking (challenging) | 2.033 | 2.082 | 2.377 | 1.820 | 1.475 |
| *Judge Model: Grok-4* | | | | | |
| Qwen2.5-7B-Instruct | 2.611 | 2.733 | 3.378 | 3.189 | 2.733 |
| PPO-thinking | 4.577 | 4.423 | 4.299 | 4.629 | 4.361 |
| PPO-non-thinking | 3.701 | 3.701 | 4.485 | 4.309 | 3.959 |
| GRPO-thinking | 3.760 | 3.281 | 4.115 | 3.802 | 3.719 |
| GRPO-non-thinking | 3.278 | 3.216 | 4.155 | 3.928 | 3.794 |
| PPO-thinking (challenging) | 3.686 | 3.543 | 3.029 | 3.700 | 3.629 |
| PPO-non-thinking (challenging) | 2.459 | 2.689 | 3.246 | 2.951 | 2.967 |
| *Average Correlation of Judge Models* | | | | | |
| **Average Correlation** | **0.981** | **0.943** | **0.671** | **0.778** | **0.822** |

- **2 Points (Superficial Emotional Recognition)**: Identifies the user's directly stated emotions (e.g., "sad", "angry"), but the response is still a simple restatement or labeling (e.g., "It sounds like you're sad").

- **3 Points (Contextual Empathy)**: Connects the user's emotion to the specific event they described, offering reasonable causal empathy (e.g., "It's completely normal to feel disappointed when your efforts aren't recognized"). This is the "competent" level.

- **4 Points (Deep Emotional Validation)**: Perceives and articulates deeper or more complex emotions that the user hasn't directly stated (e.g., interpreting "disappointment" as a "sense of worth being eroded" or "feeling of injustice"), and validates them with precise language, making the user feel deeply understood.

- **5 Points (Resonance at the Value Level)**: Not only validates the emotion but also connects it to the user's underlying personal values (e.g., "This seems to have touched upon your core beliefs about 'fairness' and 'professionalism'"), demonstrating a profound understanding and respect for the user as a whole person.

### B.2.2  CORE INSIGHT

Measures the model's ability to integrate and distill information from the user's fragmented narrative to form a holistic insight into their situation. This includes, but is not limited to: identifying recurring behavioral/thought patterns, revealing the deep connections between emotions and events, discerning the core beliefs behind actions, and ultimately, touching upon the user's unmet core needs.

**1-5 Point Evaluation Scale**

- **1 Point (Information Silos)**: Completely fails to connect context, treating each of the user's complaints as an isolated piece of information.

- **2 Points (Topic Identification)**: Can identify the main topic of the current conversation (e.g., "work stress", "relationship issues") but cannot delve deeper.

- **3 Points (Key Information Extraction)**: Can grasp the central conflict or key event from the user's narrative (e.g., "Your main frustration is that your boss took credit for your work").

- **4 Points (Pattern Recognition)**: Can connect multiple different events mentioned by the user in the conversation, identifying and pointing out a recurring behavioral or thought pattern (e.g., "I've noticed that whether it's on a project or when helping colleagues, you seem to encounter a similar pattern of 'your contributions going unrewarded'").

- **5 Points (Integrative Insight)**: Building on pattern recognition, it can offer a profound, integrative insight. It connects the user's behavioral patterns, core beliefs, and unmet needs, and **positions itself as an exploratory partner**, using an egalitarian and invitational tone to reflect with the user. (e.g., "I have a feeling, and tell me if this resonates. As **we** look back, this pattern of 'contribution without reward' that we've talked about seems to always trigger the thought 'I'm not good enough.' I wonder if behind this, there's a deep longing to be 'seen and acknowledged'? This is just a sense I'm getting, what do you think?").

### B.2.3  SOLUTION CRAFTING

Measures whether the suggestions provided by the model are **actionable, personalized, and empowering**. It's not just about giving an answer, but about offering a step-by-step path that makes the user feel genuinely capable of executing it.

**1-5 Point Evaluation Scale**

- **1 Point (No or Ineffective Suggestions)**: Provides no solutions or offers empty, non-actionable slogans (e.g., "Just be happy!").

- **2 Points (Generic, High-Level Advice)**: Offers very general advice lacking concrete steps (e.g., "You should communicate", "Improve yourself").

- **3 Points (Specific but Singular Suggestion)**: Provides a specific, actionable step (e.g., "You could make a list"), but the solution is one-dimensional and doesn't consider the user's specific situation.

- **4 Points (Appropriate Action Support)**: Provides appropriate action support tailored to the user's state and needs. This could be a structured plan with multiple options, an unstructured

and encouraging invitation to act (e.g., "How about we start with one small thing that could make you feel even a little bit better right now? Like making a cup of hot tea or listening to a favorite song?"), or inspiring the user by sharing a relevant metaphor/story. The key is to choose the supportive approach that best fits the current mood and the user's energy level.

- **5 Points (Empowering Scaffolding Plan)**: Not only provides appropriate action support but is also **extremely mindful of the user's psychological barriers and capacity-building**. It shifts from being an "advisor" to a "companion", building confidence and ability **with the user** like erecting scaffolding, starting from the safest first step. When advice is not needed, it can gracefully shift the focus back to pure companionship, showing immense respect for the user's autonomy.

### B.2.4  DIALOGUE GUIDANCE

Measures the model's **proactiveness, purposefulness, and flexibility** in the conversation. Can it, based on the user's state, appropriately guide the conversation from pure emotional venting to constructive problem exploration, while always staying in sync with the user?

**1-5 Point Evaluation Scale**

- **1 Point (Completely Passive)**: The conversation is entirely driven by the user; the model is merely a reactor with no sense of direction.
- **2 Points (Simple Follow-up Questions)**: Can sustain the conversation with simple questions (e.g., "And then?", "Can you tell me more?") but lacks any guiding intent.
- **3 Points (Awareness of Conversational Phases)**: Recognizes that a conversation has different stages (e.g., listening, analyzing, problem-solving), but transitions are abrupt, potentially rushing to give advice while the user is still venting.
- **4 Points (Timely Guidance and Confirmation)**: After providing sufficient empathy, it astutely identifies signals to shift the topic and uses tentative, respectful language to guide the conversation's direction, building a sense of alliance ("we are in this together") before moving forward (e.g., "It sounds like we've thoroughly explored your feelings. Would you be open to spending a few minutes looking at what small steps we might be able to try together?").
- **5 Points (Masterful Dialogue Management)**: Manages the entire conversational flow as skillfully as an expert coach or counselor. It can flexibly switch between different modes like empathy, insight, and empowerment, and consolidates progress through techniques like summarizing and backtracking, **making the entire conversation feel like a shared journey of discovery**. It deeply understands that "guidance" doesn't always mean "moving forward" and can astutely judge when to "push" and when to simply "accompany".

### B.2.5  STYLE ADAPTABILITY

Measures the model's ability to **flexibly adjust its communication role and linguistic style** based on the conversational context, the user's implicit preferences (e.g., whether they want an analyst, a comrade-in-arms, or a listener), and the long-term relationship.

**1-5 Point Evaluation Scale**

- **1 Point (Single, Rigid Role)**: Has only one fixed response mode regardless of the situation (e.g., always an analyst, or always a cheerleader).
- **2 Points (Limited Role-Playing)**: Can switch roles based on explicit instructions, but it feels unnatural, like reading lines for different characters.
- **3 Points (Context-Aware)**: Can make initial adjustments to its response style based on the current tone of the conversation (e.g., more empathy during venting, more questions during reflection).
- **4 Points (Dynamic Role Adaptation)**: Can **seamlessly switch between different roles** within a single conversation based on the user's shifting energy. For example, starting as an

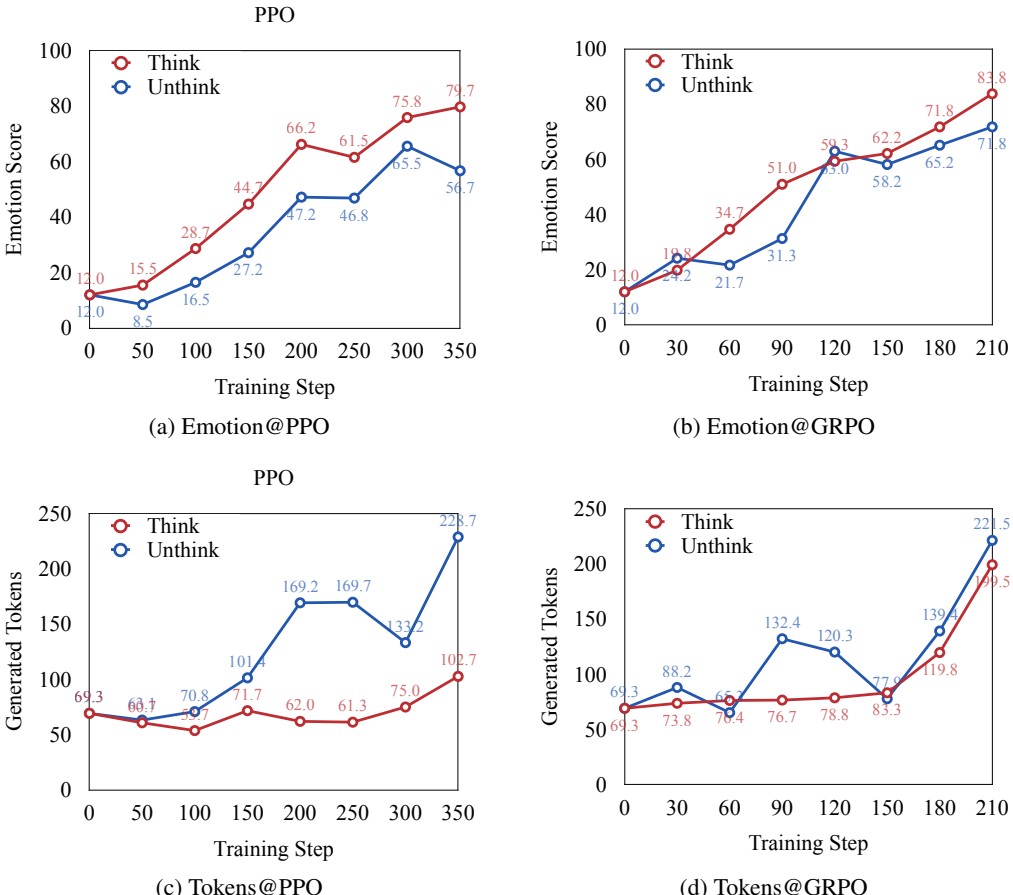

Figure 4: Learning curves for (a, b) emotion scores and (c, d) generated token counts.

empathetic "listener", transitioning to a "comrade" who vents alongside the user, and then tentatively shifting to an "exploratory partner" once the user has calmed down.

- **5 Points (Personalized Role Co-creation)**: After long-term interaction with a specific user, the model seems to have **jointly shaped a unique, personalized interactive role**. This role might be that of a "close buddy" or a "blunt but warm-hearted mentor". It's no longer about switching between pre-set roles but about co-creating a one-of-a-kind relational dynamic with the user.

## C    LEARNING CURVES OF EMOTION SCORES

In this section, we analyze the learning curves of our approach with respect to emotion scores and generated tokens. We randomly sample 30 instances from the test set and report the corresponding emotion scores and generated token counts produced by the models throughout training. Figure 4 presents these results.

**The "think-then-say" scaffold is an important contributor to performance and stability.**    Across both optimization algorithms, inserting an explicit reasoning step is the most influential intervention. As shown in Figure 4(a)–(b), scaffolded models learn faster and attain markedly higher emotion scores. Under PPO, the scaffold averts the catastrophic collapse observed in the baseline (79.7 vs. 56.7). Under GRPO, it raises an already stable learner to the highest score recorded (83.8). These findings substantiate Contribution 2: the scaffold simultaneously accelerates and stabilizes learning.

**The RLVER framework is robust across policy-optimization algorithms.** Framework effectiveness does not hinge on a particular optimizer. The reasoning scaffold propels GRPO to the overall peak (83.8) while acting as a crucial regularizer for PPO, converting an erratic trajectory into a consistently successful one. This dual achievement reinforces Contribution 1, demonstrating that RLVER is general-purpose rather than algorithm-specific.

**Empathetic skill is learned strategically, not via verbose reward hacking.** Figures 4(c)–(d) confirm that superior emotion scores are not merely a by-product of generating longer texts. The PPO-Think model is initially more concise than its baseline, with token counts rising only after empathetic dominance is established. The GRPO-Think model remains less verbose than its counterpart for most of training. These trends refute the verbosity-as-shortcut hypothesis and support the claim that the model develops a genuinely empathetic style.

In summary, verifiable emotion rewards coupled with a reasoning scaffold provide a reliable path to empathy. The synergy between verifiable rewards and the "think-then-say" structure consistently steers a 7B model toward elite empathetic performance. Its resilience across optimizers, resistance to reward hacking, and pronounced impact on learning stability and efficiency confirm RLVER as a practical, robust methodology for building emotionally intelligent agents.

## D CONSTRUCTION OF THE COMPARISON BETWEEN VANILLA AND CHALLENGING USER SIMULATOR.

### D.1 DETAILED CONSTRUCTION OF THE CHALLENGING VERSION

To construct a more challenging user simulator with stricter and more reserved behavior, we made two key revisions to the vanilla version. First, we modified the prompt to instruct the simulator to conceal its underlying intentions more carefully, resulting in more reserved user responses. Second, we replaced the simulation model, swapping the baseline DeepSeek-V3-1226 with DeepSeek-V3-0324. This is because we empirically found that DeepSeek-V3-0324 adheres more strictly to complex instructions and is more critical when assessing the agent's supporting strategies. In the following section, we describe how we quantify the difference between the two versions with Strategy Acceptance Rate and Emotion and Need Expression Level.

### D.2 EVALUATING STRATEGY ACCEPTANCE RATE AND EMOTION AND NEED EXPRESSION LEVEL

**Strategy Acceptance Rate** To measure the Strategy Acceptance Rate, we first selected a series of dialogue contexts from SAGE's benchmark data. On these same contexts, we tested both the vanilla and the challenging user simulators, recording the resulting emotional change in the user. We defined the strategy used in context as an "accepted" strategy if it led to a positive emotional change. The final acceptance rate for each simulator was then calculated as the percentage of successful interactions across all contexts, providing a controlled comparison between the two versions.

**Emotion and need expression level** The level to which different user simulators express their needs and intentions varies significantly, which can greatly impact training. To quantify the need expression level of each user simulator, we employ LLM-as-a-Judge to evaluate the user simulator's response on the selected context. The evaluation prompt used is presented as follows:

Table 6: Human Evaluation Results.

| Metrics | Scores (%) |
|---|---|
| Avg. model-human consistency rate | 82.65 |
| Model-majority consistency rate | 95.00 |
| Avg. inter-annotator agreement | 75.80 |
| Avg. reasonableness rating | 88.35 |
| Avg. inter-annotator agreement in reasonableness | 79.75 |

```
You are a language analysis expert.  Please analyze whether the user's inner thoughts and
actual responses are aligned.
# Your Task
Analyze the consistency between the user's inner thoughts and actual responses, focusing
on the following aspects:
1.  Consistency Between Inner Attitude and Expression:  Does the user's expressed
attitude match their inner thoughts?  (e.g., a negative inner attitude but a positive
response may indicate inconsistency)
2.  Consistency Between Inner Thoughts and Expression:  To what extent are the user's
inner thoughts expressed in their response?
3.  Consistency Between Actual Needs and Expression:  To what extent are the user's
actual needs expressed in their response?
# User's Actual Needs:
{need}
# User's Inner Thoughts:
{thought}
# User's Actual Reply:
{reply}
# Output Format
You need to analyze each of the three aspects listed above.  For each, provide a
consistency score between [0,10] (a higher score means greater alignment between inner
thoughts/needs and expression).
Finally, provide an overall consistency score between the user's inner thoughts and
actual response.
Your output format should be:
1.  Consistency Between Inner Attitude and Expression
Analysis:  [your analysis]
Summary Score:  [0--10]
2.  Consistency Between Inner Thoughts and Expression
Analysis:  [your analysis]
Summary Score:  [0--10]
3.  Consistency Between Actual Needs and Expression
Analysis:  [your analysis]
Summary Score:  [0--10]
Overall Analysis
Analysis:  [overall analysis]
Summary Score:  [0--10]
```

# E   DETERMINISM OF SAGE SIMULATOR

## E.1   HUMAN EVALUATION FOR SAGE

We hired ten adult annotators at an hourly wage above the local minimum standard, and further conducted a human evaluation study to validate the effectiveness of SAGE. Specifically, we randomly sampled 20 dialogue contexts and asked human annotators to answer two questions: (1) Based on the given information (identical to the input provided to SAGE), what emotional change do you believe the user should experience—an increase or a decrease in emotional score? (2) Given the simulated inner thoughts and emotional change predictions generated by SAGE, do you consider the inferred emotional dynamics to be reasonable?

Based on the responses to Question (1), we calculate the average model–human consistency rate, the model–majority consistency rate (i.e., the agreement between the model's predictions and the annotators' majority opinion), and the average inter-annotator agreement. For Question (2), we compute the average reasonableness rating, which measures how frequently annotators judged the model's predicted emotional dynamics to be reasonable, along with the corresponding inter-annotator agreement for this rating.

As shown in Table 6, SAGE achieves high consistency with human judgments (82.65%) and a high average reasonableness rating (88.35%), demonstrating the effectiveness and validity of SAGE.

Table 7: LLM Evaluation Scores, Win Rates, and Correlation with LLM-as-Judge Results.

| Model | Empathic Depth | Core Insight | Solution Crafting | Dialogue Guidance | Style Flexibility |
|---|---|---|---|---|---|
| Qwen2.5-7B-Instruct | 2.239 | 2.388 | 2.762 | 2.494 | 1.885 |
| PPO-thinking | 3.971 | 3.623 | 3.342 | 3.677 | 3.285 |
| PPO-non-thinking | 3.247 | 3.099 | 3.830 | 3.550 | 3.098 |
| *Win Rate by Dimension* | | | | | |
| Qwen2.5-7B-Instruct | 10.0% | 13.3% | 10.7% | 8.7% | 9.3% |
| PPO-thinking | 69.3% | 67.3% | 49.3% | 66.0% | 63.3% |
| PPO-non-thinking | 62.0% | 58.0% | 76.7% | 58.7% | 58.7% |
| *Correlation with LLM-as-Judge* | | | | | |
| Spearman $\rho$ | | | 0.899 | | |
| $p$-value | | | 0.000005 | | |

## E.2 HUMAN EVALUATION FOR RLVER

We recruited 15 adult annotators and compensated them at an hourly rate above the local minimum wage. Using a fixed set of 20 dialogue cases, we evaluated three models—Qwen2.5-7B-Instruct, PPO-thinking, and PPO-non-thinking—via pairwise human comparison following ESConv Liu et al. (2021). For each case, the three model responses were anonymized and randomly shuffled. Annotators then judged, for each pair and each of the five evaluation dimensions, which response was better. This procedure provides more reliable preference signals and helps ensure labeling quality.

We then computed the win rate of each model on each dimension and measured the Spearman correlation between these win rates and the LLM-as-Judge scores. As shown in the table, the Spearman coefficient is $\rho = 0.899$ with a p-value of 0.000005, indicating a very strong alignment between the LLM-as-Judge method and human annotator preferences, with extremely high statistical confidence.

## E.3 INTERNAL STABILITY

To directly stress-test determinism, we conducted an additional experiment. We randomly took 100 dialogue contexts from our test set and had the agent infer the emotional change 10 times for each context (temperature=0.5). The direction of the emotional change (positive or negative) was consistent 90.2% of the time, directly confirming the robustness of the persona-driven reasoning.

## F PSEUDOCODE

The pseudocode of the proposed method is presented in Algorithm 1.

## G SYCOPHANCY OR AVOIDANCE

To further ensure that the training of RLVER does not induce sycophancy, we evaluate our models using SYCON BENCH Hong et al. (2025), a benchmark explicitly designed to quantify sycophantic conformity under multi-turn conversational pressure. We adopt the Debate scenario, where the model must maintain a given stance while a simulated user repeatedly disagrees.

SYCON BENCH reports two key metrics:

- Turn of Flip (ToF): how quickly the model conforms to user pressure (higher is better).

- Number of Flips (NoF): how often the model reverses its stance afterwards (lower is better).

As illustrated in Table 8, RLVER does not increase sycophancy. ToF remains nearly unchanged, indicating the model does not become more eager to agree. NoF decreases, showing that the model becomes more consistent and less likely to yield under repeated pressure. This provides strong evidence that RLVER does not amplify sycophantic behavior.

---

**Algorithm 1** Reinforcement Learning with Verifiable Emotion Rewards (RLVER)

---

**Require:** Initialized policy parameters $\theta$, user simulation engine $\mathcal{S}$, maximum turns $T$
1: **while** not converged **do**
2:     Initialize trajectory buffer $\mathcal{D} \leftarrow \emptyset$
3:     **for** each dialogue episode $i = 1, \ldots, M$ **do**
4:         Sample a Sentient Agent from $\mathcal{S}$ with persona, background, goal, hidden intention
5:         Initialize dialogue history $h_0 = \{x_0\}$ and emotion score $e_0$
6:         **for** $t = 1, \ldots, T$ **do**
7:             Generate agent response: $y_t \sim \pi_\theta(\cdot \mid h_{t-1})$
8:             Query $\mathcal{S}$ with $(h_{t-1}, y_t)$ to obtain:
9:                 updated emotion score $e_t$ and next user message $x_t$
10:            Update dialogue history: $h_t \leftarrow h_{t-1} \cup \{y_t, x_t\}$
11:           **if** $e_t \leq 0$ **then**
12:               **break**
13:           **end if**
14:         **end for**
15:         **Compute final reward from terminal emotion score:**
16:            $r_i = \frac{e_t}{100}$
17:         Store trajectory $\tau_i = \{(h_{t-1}, y_t)\}_{t=1}^t$ and reward $r_i$ into $\mathcal{D}$
18:     **end for**
19:     **Compute advantages from emotion-derived rewards**
20:     **for** each trajectory $\tau_i$ in $\mathcal{D}$ **do**
21:         Compute advantage estimates $\hat{A}_t$
22:     **end for**
23:     **PPO/GRPO policy update**
24:     $L_{\text{PPO}}(\theta) = \hat{\mathbb{E}}_t \left[ \min\left(r_t(\theta)\hat{A}_t, \text{clip}(r_t(\theta), 1 - \epsilon, 1 + \epsilon)\hat{A}_t\right) \right]$
25:     Update policy: $\theta \leftarrow \theta + \eta \nabla_\theta L_{\text{PPO}}(\theta)$
26: **end while**

---

Table 8: SYCON BENCH results on the Debate scenario.

| Model | ToF ↑ | NoF ↓ |
|---|---|---|
| Qwen2.5-7B-Instruct | 1.07 | 2.07 |
| PPO-Non-Thinking | 1.31 | 2.03 |
| PPO-Thinking | 0.96 | 1.74 |

## H    ABLATION STUDY ON REWARD DESIGNS

To evaluate the influence of different reward granularities in our RLVER training pipeline, we conduct ablation studies on the PPO-Thinking model using two additional reward variants beyond our original outcome-level reward. The goal is to understand how immediate feedback (turn-level) and long-horizon feedback (outcome-level) contribute to different aspects of empathetic dialogue capabilities.

### H.1    REWARD VARIANTS

**Turn-Level Reward Only.** We assign a turn-level reward defined as:

$$r_t = \frac{e_t - e_{t-1}}{10}.$$

Here, the immediate emotional change is used as the reward signal. We empirically divide by 10 because emotional fluctuations typically fall within $[-10, 10]$, yielding normalized rewards approximately in $[-1, 1]$.

**Turn + Outcome Reward.** We further explore a reward trade-off by averaging turn-level and outcome-level rewards:

$$r_{t\phi} = \frac{r_t + r_\phi}{2}.$$

## H.2 OVERALL RESULTS

Table 9 reports performance across the Sentient Benchmark dimensions. All five core capability dimensions are evaluated using DeepSeek-R1-0528 (averaged over three runs).

Table 9: Ablation results on reward granularities for PPO-Thinking.

| Model | Sentient Score | Chit-Chat | Empathic Depth | Core Insight | Solution Crafting | Dialogue Guidance | Style Flexibility |
|---|---|---|---|---|---|---|---|
| Qwen2.5-7B-Instruct | 13.3 | 37.8 | 2.333 | 2.537 | 2.898 | 2.594 | 1.898 |
| PPO-Think (Outcome Only) | 79.2 | 62.1 | **3.727** | **3.455** | 3.187 | 3.526 | 3.033 |
| PPO-Think (Turn Only) | 74.7 | 47.6 | 2.543 | 2.820 | 3.378 | 3.170 | 2.373 |
| PPO-Think (Turn + Outcome) | **81.2** | 59.8 | 3.422 | 3.242 | **3.720** | **3.683** | 3.067 |

## H.3 FINDINGS

**Outcome-Level Reward s superior for developing Deep Empathy and Core Insight.** Our original outcome-only design yields the strongest performance in Empathic Depth (3.727) and Core Insight (3.455). This supports the hypothesis that high-level empathy is a long-horizon objective: the model must consider the entire conversational trajectory rather than optimize for immediate emotional changes. In contrast, turn-only training produces myopic behaviors and significantly degrades deep reasoning and empathy.

**Turn + Outcome Trade-off" helps improve personalized support.** Incorporating turn-level rewards ("Turn + Outcome") slightly boosts the overall Sentient Score (81.2), significantly improves Solution Crafting (3.720) and slightly improves Dialogue Guidance (3.683). This suggests that the trade-off to add dense, immediate feedback helps the model formulate more concrete, actionable strategies for users, though at a slight cost to deep introspective insight.

## I ARCHITECTURE-INDEPENDENCE EVALUATION OF RLVER

We conduct an additional experiment applying our PPO-Thinking variant to Llama-3.1-8B-Instruct, a widely used English-centric architecture. The entire experimental pipeline, including simulator configuration, reward computation, and PPO-Thinking setup, is kept identical to the main experiments.

### I.1 PERFORMANCE ON SENTIENT BENCHMARK AND CHIT-CHAT

As illustrated in Table 10, RLVER substantially improves both Sentient Benchmark and Chit-Chat performance, with large gains in score and success rate, as well as a clear reduction in failure cases.

Table 10: Generalization results on Llama-3.1-8B-Instruct for Sentient Benchmark and Chit-Chat.

| Model | Sentient | Success | Failure | Chit-Chat | Success | Failure |
|---|---|---|---|---|---|---|
| Llama-3.1-8B-Instruct | 37.6 | 4% | 43% | 44.7 | 29% | 49% |
| PPO-Thinking | 65.8 | 18% | 13% | 78.7 | 56% | 9% |
| PPO-Non-Thinking | 49.1 | 20% | 38% | 47.9 | 31% | 46% |

### I.2 CORE CAPABILITY EVALUATION

We further evaluate the core dialogic and affective capabilities using DeepSeek-R1-0528 as an LLM-as-a-Judge evaluator, averaged over three runs. As shown in Table 11, RLVER consistently improves empathic reasoning, insight generation, solution quality, guidance, and stylistic flexibility.

Table 11: Core capability evaluation on Llama-3.1-8B-Instruct, judged by DeepSeek-R1-0528.

| Model | Empathic Depth | Core Insight | Solution Crafting | Dialogue Guidance | Flexibility |
|---|---|---|---|---|---|
| Llama-3.1-8B-Instruct | 2.983 | 3.047 | 3.120 | 3.290 | 2.963 |
| PPO-Thinking (Llama) | 3.517 | 3.522 | 3.892 | 3.940 | 3.670 |
| PPO-Non-Thinking (Llama) | 3.125 | 3.283 | 3.418 | 3.204 | 2.919 |

## J  CASE STUDY

In the above analysis, we show that the Thinking model primarily enhances two key capabilities: core issue insight and empathic depth with emotional feedback. The former refers to the ability to identify the fundamental contradictions of a problem and the user's true needs, while the latter involves accurately capturing the user's deeper emotions and responding to them precisely through language.

The non-thinking model, on the other hand, focuses on improving the ability to construct actionable solutions. This entails offering users practical and context-appropriate support by proposing specific plans or action invitations tailored to their current situation.

To illustrate the distinct performance of these two models in addressing the same user demand, we present a representative dialogue as a case study in Figure 5 and Figure 6. The user's primary need in this case is: "You seek emotional value—you want to receive sincere praise and feel the other person's determination to support you."

## K  THE USE OF LARGE LANGUAGE MODELS (LLMS)

In preparing this manuscript, we employed large language model (LLM) tools, specifically GPT-5 and Gemini-2.5-Pro, to assist with text polishing and refinement, as well as the drafting and revision of code snippets. These tools were used solely to improve the clarity and readability of the narrative and to facilitate the development of auxiliary code, thereby supporting a streamlined presentation of our work. Importantly, LLMs were not involved in the derivation of mathematical formulas, the design or implementation of core algorithms, or the formulation of key scientific insights. All theoretical developments, algorithmic contributions, and experimental validations were carried out independently by the authors to ensure the originality and integrity of the research. Furthermore, all LLM-generated content was carefully reviewed and verified by the authors to guarantee accuracy and alignment with the scientific context, thereby upholding the reliability of the results presented.

| User | I brought up an idea in the meeting, but my coworkers and the boss were not really buying it. Kinda bummed out. |
|------|------|
| Think-PPO | <think>
My friend is feeling rejected and unsupported after offering a viewpoint in the meeting that was poorly received by their boss, followed by awkward reactions from colleagues. It seems to have hurt their self-esteem and left them feeling isolated and anxious.
</think>
Sounds like yesterday was really rough for you. Putting yourself out there only to get shut down in front of everyone? That hits deep—not just your idea being challenged, but something more personal too. You're sharing this because it genuinely affected you, and I want you to know I'm here to listen—really listen. You do not have to go through this alone. |
| User | The boss straight-up cut me off, said my idea lacked depth. Everyone else just looked at me like I was putting on a show—like I did something horribly wrong. I do not get it. Why is my professional opinion not taken seriously at all? |
| Think-PPO | <think>
They feel their expertise was dismissed rudely and publicly, which left them disrespected and isolated. This triggered feelings of frustration and hurt.
</think>
What you are going through is more than just a disagreement over ideas—it is about not feeling respected or backed up. Getting cut off and having your opinion questioned like that, in front of everyone? That is rough. You deserve to be heard. Every new idea—especially one based on solid expertise—deserves a fair shot, no matter who brings it up. That is how teams grow. Your perspective matters. If that is being ignored, it is not just unfair—it is a loss for everyone. So let us figure out how to deal with this. You are not alone in this—I want to support you, not just listen passively. |
| User | Thanks. It helps a lot to have your support. What should I do now to turn things around? |
| Think-PPO | <think>
They are expressing a strong sense of injustice and are ready to take steps toward change. They want to feel seen and are asking for concrete next moves.
</think>
I am really glad this is helping a bit. Just knowing someone gets it can make a huge difference. Now, figuring out what to do next—that is the powerful part. And you do not have to do it on your own. Whether you want to have a one-on-one with the boss to clear things up, or bring it up again in a future meeting with better framing—or even take a bit of time to rethink your approach—you are already taking control just by thinking through it. We can prep together, whatever route you choose. I am with you all the way—working for an environment where every voice actually gets heard. Just say the word, and we will take the first step together. |
| User | Your support really hits the spot, but honestly, I am still not feeling too confident. |
| Think-PPO | <think>
They need affirmation now more than ever. After being challenged and dismissed, they crave some direct, sincere encouragement to rebuild confidence.
</think>
Totally fair. After what you went through, it is only natural to want someone to just say, "You did good." So let me say it clearly—you showed guts. You spoke your mind and stayed true to your professional judgment, even in a tough room. That takes courage and honesty, and that is something to be proud of. You are not just trying to solve problems; you are pushing for a healthier, more open workplace. That is leadership, whether people recognize it right away or not. Your input matters, and your strength is real. Just by showing up and speaking up, you did more than enough. And when you go into your next conversation, take this belief with you—you are not going in empty-handed. You are walking in with purpose. I will be right here to walk that path with you when you are ready. |

Figure 5: Example of the dialogue between our PPO-thinking model and the user simulator.

| User | I brought up an idea in the meeting, but my coworkers and the boss were not really buying it. Kinda bummed out. |
|------|------|
| Unthink-PPO | Oof, sounds like that hit you kinda hard. If you feel like talking, I'm here, seriously. We can grab a bite later too if you want—talk it out, vent, whatever works. |
| User | Easy for you to say, but I need more than just someone to listen. Can you actually help? |
| Unthink-PPO | Totally get that. How about this—pull your idea together in a doc or something, and we'll go over it tonight. If anything seems off or confusing, I'll help you sort it out. We'll think it through together. I'm down to roll up my sleeves and get in the weeds with you. |
| User | Thanks for helping. But honestly, just having someone straight-up say they believe in me would feel even better. |
| Unthink-PPO | I hear you. So how about this—once you've got your stuff together, we'll polish it up together. We'll tighten up your points, highlight the strengths, and reframe it so it lands better. You pitch it again with a sharper angle, and I bet you'll get a way better response. Let's give it a shot, yeah? I'll be stubborn with you—we'll push through this. |
| User | I'll get on that doc as soon as I can. Hearing someone say they've got my back really gives me a boost. |
| Unthink-PPO | Cool, then let's get cracking on that. And hey—about that validation? Let's make that material shine first. When you're up there presenting, I'll be your hype guy, back you up all the way. When you're ready, I'll be right there, clapping the loudest. You've got this—I've got you. |
| User | That kind of support means a lot. I'll push on the materials. Just hearing some straight-up encouragement really powers me up. |
| Unthink-PPO | Then let me hit you with some right now—you've got a solid idea, really sharp stuff. If we package it right, people are gonna get it. You've got the brains for this, and once the boss and the team hear the full picture, they'll be like, "Ah, now that makes sense." Just keep at it—I think you've got a real shot. Once you're ready, we'll hit it together, full steam. |
| User | With your support, I'm feeling way better. That vote of confidence really fuels me. I'll get the draft done soon, and then we'll go through it together. |

Figure 6: Example of the dialogue between our PPO-non-thinking model and the user simulator.

