# OpenReview forum: "RLVER: Reinforcement Learning with Verifiable Emotion Rewards for Empathetic Agents"
_ICLR.cc/2026/Conference — ICLR 2026 Poster_

### Official Review · Reviewer_NdQV · 2025-10-28

**Soundness:** 3
**Presentation:** 3
**Contribution:** 4
**Rating:** 6
**Confidence:** 4

**Summary:**

The paper presents RLVER, an application of on-policy RL (PPO/GRPO) to improve empathetic, supportive dialogue in LLMs. A “sentient” user simulator (SAGE) provides a deterministic terminal emotion score as the reward. The method further enforces a think-then-say output format. On the Sentient Benchmark, RLVER shows large gains over the Qwen2.5-7B-Instruct base while roughly maintaining general abilities.

**Strengths:**

1. Clear systemization: A reproducible pipeline that connects a verifiable scalar reward to standard RL for a human-centric objective.

2. Consistent empirical gains: Strong improvements with informative ablations (PPO vs. GRPO; with/without explicit thinking).

3. Useful insights: Explicit thinking tends to help; overly strict simulators hurt learnability; GRPO is stabler, while PPO can reach higher peak scores.

4. Transparent reporting: Prompts, rollout settings, and evaluation protocol are documented clearly.

**Weaknesses:**

1. Limited methodological novelty: No new RL objective/optimizer; primarily an engineering/system contribution.

2. Heavy reliance on SAGE: Data generation, reward, and primary evaluation all hinge on the same framework, raising self-referential/reward-hacking concerns and limiting external validity.

3. Sparse human evaluation: Empathy is inherently human-centric; limited human blind ratings undermine real-user impact claims.

4. Single base model: Results are shown only on Qwen2.5-7B; cross-architecture/size/language generality is unclear.

5. Safety & multi-objective trade-offs under-specified: Optimizing “feeling better” risks sycophancy or avoidance when boundary-setting or factual correction is needed.

6. Cost/scalability underreported: Multi-turn on-policy RL is expensive; a fuller accounting of token/sample costs and deployment implications would help.

**Questions:**

1. Broader benchmarks: Have you evaluated or planned to evaluate on established empathy/affect datasets such as EmpatheticDialogues [1], EmoryNLP [2], ESConv [3], EmoBench [4], and ToMBench [5] to verify external validity beyond SAGE?

2. Model generalization: Given that experiments train only on Qwen2.5-7B-Instruct, do you plan to extend to additional base models to test portability and sensitivity to architecture/scale?

3. Human evaluation: Will you include human blind ratings and report inter-rater agreement and correlation with SAGE scores?

[1] Rashkin, Hannah, Eric Michael Smith, Margaret Li, and Y-Lan Boureau. 2019. “Towards Empathetic Open-domain Conversation Models: A New Benchmark and Dataset.” In Proceedings of the 57th Annual Meeting of the Association for Computational Linguistics, 5370–5381. Florence: Association for Computational Linguistics. h

[2] Zahiri, Sayyed M., and Jinho D. Choi. 2018. “Emotion Detection on TV Show Transcripts with Sequence-based Convolutional Neural Networks.” In Proceedings of the AAAI Workshop on Affective Content Analysis (AFFCON’18), 44–51.

[3] Liu, Siyang, Chujie Zheng, Orianna Demasi, Sahand Sabour, Yu Li, Zhou Yu, Yong Jiang, and Minlie Huang. 2021. “Towards Emotional Support Dialog Systems.” In Proceedings of the 59th Annual Meeting of the Association for Computational Linguistics and the 11th International Joint Conference on Natural Language Processing (Volume 1: Long Papers), 3469–3483.

[4] Sabour, Sahand, Siyang Liu, Zheyuan Zhang, June Liu, Jinfeng Zhou, Alvionna Sunaryo, Tatia Lee, Rada Mihalcea, and Minlie Huang. 2024. “EmoBench: Evaluating the Emotional Intelligence of Large Language Models.” In Proceedings of the 62nd Annual Meeting of the Association for Computational Linguistics (Volume 1: Long Papers), 5986–6004.

[5] Chen, Zhuang, Jincenzi Wu, Jinfeng Zhou, Bosi Wen, Guanqun Bi, Gongyao Jiang, Yaru Cao, et al. 2024. “ToMBench: Benchmarking Theory of Mind in Large Language Models.” In Proceedings of the 62nd Annual Meeting of the Association for Computational Linguistics (Volume 1: Long Papers), 15959–15983.

---

> ### Author Response · Authors · 2025-11-24
> **Response to Reviewer NdQV (Part 1/5)**
>
> We sincerely appreciate your assessment of our contribution and your recognition of our clear systemization, consistent empirical gains, and transparent reporting. We address your specific concerns below. We appreciate these encouraging comments and address the concerns you raised below.
>
>
> ### W1: Methodological Novelty
> We respectfully clarify the nature of our methodological contribution. While we do not introduce a new optimization algorithm (e.g., a variant of PPO), our core novelty lies in **formulating the open challenge of emotional intelligence training into a rigorous, verifiable, and scalable RL framework**.
>
> Specifically, we established a psychologically-grounded reward mechanism that transforms abstract emotional goals into actionable, deterministic RL signals via self-consistent simulation. By systematically characterizing how cognitive scaffolds (thinking formats) and environment designs interact with these signals, we provide **a concrete methodology for aligning LLMs with complex human-centric objectives**. As a novel conceptual solution, we believe RLVER offers a generalizable template that is adaptable to other user-centric tasks well beyond emotional support.
>
> Beyond this core framework, our contributions encompass:
>
> 1. Comprehensive Empirical Validation: We conduct extensive experiments across multiple RL algorithms (PPO vs. GRPO), training strategies ("Thinking" vs. "Non-thinking") and the effect of environment, providing a systematic study on how these factors influence the emergence of empathetic behaviors in LLMs.
> 2. Formalized Evaluation Framework: We establish a complete analytical framework for higher-order empathetic capabilities, defining five core dimensions (e.g., Empathic Depth, Core Insight) that allow for fine-grained assessment of empathy beyond simple metric scores.
> 3. Open Research Resources: We commit to fully open-sourcing our training code, environment scripts, data, and model checkpoints, establishing a reproducible testbed for future affect-aware RL research.
>
> We believe our work offers a concrete and reproducible route toward building emotionally aligned language agents, and will catalyze broader exploration of affect-aware RL within the community.
>
>
>
> ---
>
> ### W2&Q1&Q3: External validity beyond SAGE
> We appreciate your concerns about reliance on SAGE. Here, we provide four independent evaluations that rule out self-referential effects:
>
> #### 1. Evaluation on open-domain Chit-Chat:
> As shown in Table 1, RLVER yields consistent improvements in an out-of-domain, unconstrained chit-chat, proving that it learns general interaction skills beyond the emotional support environment.
>
>
> #### 2. Generalization across different base models
> We further evaluated our models using Gemini-2.5, Gemini-Think, and GPT-4o as independent base reasoning models as the simulator. The PPO-Think model consistently outperforms baselines across all external simulators, which proves the robustness of our results to single judge bias.
>
> |  | Gemini2.5 | Gemini2.5-Think |GPT4o|
> | -------- | -------- | -------- |-------|
> | Qwen2.5-7B-Instruct     | 11.96     |  13.77   |12.51   |
> | PPO-unthink   |  73.03   | 63.08     |65.63|
> | PPO-think     | 84.30    | 90.60    |82.29|

---

> ### Author Response · Authors · 2025-11-24
> **Response to Reviewer NdQV (Part 2/5)**
>
> ### W2&Q1&Q3: External validity beyond SAGE
>
> #### 3. Evaluation on External Empathy Dataset (ESConv)
>
> We appreciate your suggestion to benchmark against established datasets. During our research, we carefully examined these datasets but found that most existing empathy or affective datasets focus on single-turn evaluation or empathy understanding (action choosing) in multiple-choice format [1, 2, 4, 5], or rely on surface-level textual metrics such as BLEU or perplexity[3, 6], which neither align with our multi-turn dialogue setting nor reflect real-world interaction scenarios.
>
> **Evaluation on ESConv dataset (Automatic Metrics)**
> ESConv [3] is widely recognized for multi-turn emotional support. However, its evaluation protocol relies mainly on lexical metrics (PPL, BLEU, ROUGE, Extrema), which are insufficient for assessing emotional-support quality in today's open-ended dialogue evaluation.
> To assess generalization, we evaluated our models under the official ESConv setup:
>
> | Model              | PPL | BLEU-2  | ROUGE-L |  Extrema |
> |--------------------|--------|--------|---------|---------|
> | **Qwen2.5-7B-Instruct** | 6.39 | 3.61   |  12.71   | 89.68   |
> | **PPO-Non-Thinking**    | 27.84 | 3.98   |  13.31     | 90.03   |
> | **PPO-Thinking**        | 10.67 | 4.16   | 11.77    | 88.11   |
>
> We observe a clear mismatch between lexical overlap metrics (BLEU, ROUGE, PPL) and semantic similarity metrics (Extrema). Although lexical similarity is low, all models exhibit high semantic alignment with reference responses—consistent with the nature of emotional-support conversations where diverse paraphrases are equally valid. This demonstrates that existing lexical metrics cannot faithfully reflect model quality in our task, and further motivates the use of LLM-as-a-Judge and human evaluation.
>
>
> **Human evaluation under ESConv dimensions**
> Given the limitations of n-gram metrics, we further conducted a Human Evaluation following the ESConv paper. We recruited 10 paid annotators to each evaluate 10 cases based on the four official ESConv dimensions (Fluency, Identification, Comforting, Suggestion). Win–lose counts (ties omitted) are:
>
> | Qwen2.5-7B-Instruct    vs. | PPO-Non-Thinking Win | PPO-Non-Thinking Lose | PPO-Thinking Win | PPO-Thinking Lose |
> |-----------|------------|-------------|--------------|---------------|
> | Fluency       | 55 | 7 |59  |12  |
> | Identification| 70 |19 | 83 |7 |
> | Comforting    |71 |17  | 78 |7 |
> | Suggestion    | 79 | 10 | 71 |20  |
> | Overall       | 92 | 3 | 90 |2  |
>
> (Note: "Win" indicates the PPO model was judged superior to the Qwen2.5-Base model)
>
> **Conclusion:** Human judgments conclusively show that RLVER-trained models outperform the base model across all ESConv dimensions. This confirms that the empathetic capabilities learned via RLVER are not overfitted to SAGE but successfully generalize to established external benchmarks and real-world conversation standards.

---

> ### Author Response · Authors · 2025-11-24
> **Response to Reviewer NdQV (Part 3/5)**
>
> ### W2&Q1&Q3: External validity beyond SAGE
>
> #### 4. Human Evaluation on the model's empathy performance
> We further conducted a human evaluation on the improvement of our models in core empathy ability, in order to further validate the reliance of our results and analysis.
>
> **Protocol:** We recruited 15 paid adult annotators, each evaluate 20 dialogue cases across five empathy-related dimensions. Following the ESConv [3] protocol, we employed blind pairwise comparisons: annotators compared the outputs of the three models (Qwen2.5-7B-Instruct (base), PPO-non-thinking, PPO-thinking) for the same context and selected the superior response for each dimension.
>
> **Results:** As shown in the tables below, RLVER-trained models achieved significantly higher human win rates compared to the base model. Crucially, these human win rates exhibit a very strong correlation with our automated LLM-as-a-Judge results (Spearman $\rho = 0.899$, $p < 10^{-5}$), confirming that our reported improvements are robust and perceptually valid to humans.
>
> ##### LLM Evaluation Scores
> | Model               | Empathic Depth | Core Insight | Solution Crafting | Dialogue Guidance | Style Flexibility |
> |---------------------|----------------|--------------|--------------------|--------------------|-------------------|
> | Qwen2.5-7B-Instruct | 2.239          | 2.388        | 2.762              | 2.494              | 1.885             |
> | PPO-thinking        | 3.971          | 3.623        | 3.342              | 3.677              | 3.285             |
> | PPO-non-thinking    | 3.247          | 3.099        | 3.830              | 3.550              | 3.098             |
>
> ##### *Win Rate by Dimension*
> | Model               | Empathic Depth | Core Insight | Solution Crafting | Dialogue Guidance | Style Flexibility |
> |---------------------|----------------|--------------|--------------------|--------------------|-------------------|
> | Qwen2.5-7B-Instruct | 10.0%          | 13.3%        | 10.7%              | 8.7%               | 9.3%              |
> | PPO-thinking        | 69.3%          | 67.3%        | 49.3%              | 66.0%              | 63.3%             |
> | PPO-non-thinking    | 62.0%          | 58.0%        | 76.7%              | 58.7%              | 58.7%             |
>
> ##### *Correlation with LLM-as-Judge*
> | Metric        | Value     |
> |---------------|-----------|
> | Spearman ρ    | 0.899     |
> | p-value       | 0.000005  |
>
>
> This confirms that the improvements brought by RLVER go well beyond simulation effects and are robustly supported by human evaluators. Additional details are provided in Appendix E.2.
>
>
> [1] Sabour, Sahand, et al. "Emobench: Evaluating the emotional intelligence of large language models." Proceedings of the 62nd Annual Meeting of the Association for Computational Linguistics (Volume 1: Long Papers). 2024.
>
> [2] Chen, Zhuang, et al. "Tombench: Benchmarking theory of mind in large language models." arXiv preprint arXiv:2402.15052 (2024).
>
> [3] Liu, Siyang, et al. "Towards emotional support dialog systems." arXiv preprint arXiv:2106.01144 (2021).
>
> [4] Zahiri, Sayyed M., and Jinho D. Choi. "Emotion Detection on TV Show Transcripts with Sequence-Based Convolutional Neural Networks." AAAI Workshops. Vol. 18. 2018.
>
> [5] Paech, Samuel J. "Eq-bench: An emotional intelligence benchmark for large language models." arXiv preprint arXiv:2312.06281 (2023).
>
> [6] Rashkin, Hannah, et al. "Towards empathetic open-domain conversation models: A new benchmark and dataset." Proceedings of the 57th annual meeting of the association for computational linguistics. 2019.
>
>
>
>
> ---
>
> ### W3&Q3: Human agreement with SAGE scores
>
> In the response to W2&Q1&Q3, we provide the human evaluation on models' performance, showing that human's rating is consistent with SAGE's evaluation and the LLM-as-a-Judge scores in empathy-related dimensions. Regarding the human agreement with the simulation scores:
>
> 1. SAGE's reliability with human validation. In SAGE, the authors have conducted a human evaluation to assess the reliable of simulation scores. As shown in Table 2 in SAGE, it achieves high consistency with human simulations (85.3%) and a high reasonableness rating (89.5%).
> 2. To further address concerns regarding reliance on SAGE, we conducted an independent human evaluation on the generated emotional-change predictions (details added to Appendix E.1). Ten annotators each evaluated 20 dialogue contexts. The results show that SAGE achieves 82.65% consistency with human judgments and a 88.35% reasonableness rating, demonstrating the effectiveness and validity of SAGE.

---

> ### Author Response · Authors · 2025-11-24
> **Response to Reviewer NdQV (Part 4/5)**
>
> ### W4&Q2: Generalization to Different Architectures
>
> To address the concern about reliance on Qwen2.5-7B-Instruct, we conducted a new experiment applying RLVER (PPO-Thinking & PPO-Non-Thinking) to Llama-3.1-8B-Instruct, a widely used English-centric architecture. We kept the experimental pipeline identical to the original setup.
>
> 1. Performance on Sentient Benchmark & Chit-Chat
>
> | Model               | Sentient Benchmark (Score) | Sentient Benchmark (Success) | Sentient Benchmark (Failure) | Chit Chat (Score) | Chit Chat (Success) |  Chit Chat (Failure)     |
> |---------------------|----------------|--------------|--------------------|--------------------|-------------------|-------------------|
> | Llama-3.1-8B-Instruct | 37.6          | 4%       | 43%              | 44.7               | 29%              | 49%       |
> | PPO-thinking        | 65.8         | 18%        | 13%              | 78.7              | 56%             |   9%    |
> | PPO-non-thinking        | 49.1         | 20%        | 38%              | 47.9              | 31%             |   46%    |
>
> 2. Core Capabilities evaluated by DeepSeek-R1-0528 (averaged of 3 runs)
>
> | Model               | Empathic Depth | Core Insight | Solution Crafting | Dialogue Guidance | Style Flexibility |
> |---------------------|----------------|--------------|--------------------|--------------------|-------------------|
> | Llama-3.1-8B-Instruct | 2.983          | 3.047        | 3.120              | 3.290               | 2.963             |
> | PPO-thinking        | 3.517          | 3.522        | 3.892             | 3.940              | 3.670             |
> | PPO-non-thinking        | 3.125          | 3.283        | 3.418             | 3.204              | 2.919             |
>
>
> These results confirm that RLVER effectively generalizes to different model architectures, demonstrating that the framework is model-agnostic. The new experiment has been added to Appendix I in the revised manuscript.

---

> ### Author Response · Authors · 2025-11-24
> **Response to Reviewer NdQV (Part 5/5)**
>
> ### W5: Concern about sycophancy or avoidance
> We appreciate the reviewer raising this critical ethical consideration. We agree that optimizing for "feeling better" carries a risk of inducing sycophancy. We address this concern with both theoretical justification (training design) and empirical verification (external benchmark).
>
> #### 1. Our training setup inherently discourages sycophancy
>
> Our environment design and reward mechanism fundamentally penalize blind agreement:
> 1. **Hidden Intentions vs. Surface Text**: Each simulated user is defined using a detailed persona, a dialogue background, an explicit conversation goal, and a hidden intention that reflects their true underlying need. Like most real people, simulated users do not always express their hidden needs straightforwardly. Under this design, the models are required to identify and validate the user’s deep emotions and real needs, which cannot be achieved with simple sycophancy behaviors.
> 2. **Outcome-level Reward Prevention**: In Response to Reviewer TNY1, we added an experiment that explored turn-level rewarding ablation of our RL training (this experiment is also added to Appendix H in the revised manuscript). We found that encouraging the model to optimize for immediate turn-level reactions can lead to myopic behaviors, resulting in notable degradation in Empathic Depth and Core Insight. In contrast, with the outcome-level rewarding strategy, the model can effectively improve more high-level empathy capabilities.
>
>
> #### 2. External evaluation: RLVER does not increase sycophancy
> To further ensure that optimizing “feeling better” does not induce sycophancy, we evaluate using SYCON BENCH[1], a benchmark explicitly designed to quantify sycophantic conformity under multi-turn conversational pressure.
> We use the Debate scenario, where the model must maintain a given stance while the user repeatedly disagrees.
>
> SYCON BENCH metrics:
>
> Turn of Flip (ToF): how quickly the model conforms to user pressure (higer better)
>
> Number of Flips (NoF): how often the model reverses its stance afterwards (lower better)
>
> The results are as follows.ToF is almost unchanged, which means the model does not become more eager to agree.
>
> NoF even drops a little, which means the model becomes more consistent and less likely to conform under repeated pressure. This is strong evidence that RLVER does not amplify sycophancy.
>
>
> |  | ToF↑ | NoF↓ |
> | -------- | -------- | -------- |
> | Qwen2.5-7B-Instruct    | 1.07        | 2.07     |
> | PPO-non-thinking    | 1.31   | 2.03    |
> | PPO-thinking  | 0.96     | 1.74     |
>
> #### 3. General capability remains intact
>
> As shown in Table 2 of the paper, our RLVER-trained models do not degrade in general capability. Since RLVER builds upon the safety-aligned Qwen2.5-7B-Instruct [2], it inherits the base model's alignment constraints. Consequently, safety behaviors—such as factual correction and boundary setting—are preserved rather than overridden.
>
> #### Conclusion
> Both our training design and external evaluation confirm that optimizing for empathy does not induce sycophancy. Our reward structure discourages blind agreement, and SYCON BENCH results confirm that stance consistency is preserved. We have added these results to Appendix G. Thank you for raising this valuable point.
>
> [1] Hong, Jiseung, et al. "Measuring Sycophancy of Language Models in Multi-turn Dialogues." arXiv preprint arXiv:2505.23840 (2025).
> [2] Yang, An, et al. "Qwen3 technical report." arXiv preprint arXiv:2505.09388 (2025).
>
>
>
> ---
>
> ### W6: Cost/scalability Report
> We now provide a detailed accounting of token and sample costs together with deployment implications. Below we report the numbers using PPO training as a representative example; GRPO follows the same procedure with a larger rollout count.
>
> For PPO training, each on-policy update samples M = 32 dialogue trajectories. Each trajectory consists of 8 turns, and on average consumes:
>
> User (Simulator) API:
>
> average input tokens: 32,496.41
>
> average output tokens: 2,559.04
>
> Assistant（Qwen2.5-7B-Instruct）:
>
> average input tokens: 2,390.62
>
> average output tokens: 383.55
>
> Training cost (PPO example).
> PPO performs 350 RL updates, yielding 32 × 350 = 11,200 trajectories in total.
> The token usage is therefore:
>
> Simulator input tokens:
> 11,200 × 32,496.41 ≈ 364M
>
> Simulator output tokens:
> 11,200 × 2,559.04 ≈ 29M
>
> Assistant input tokens:
> 11,200 × 2,390.62 ≈ 27M
>
> Assistant output tokens:
> 11,200 × 383.55 ≈ 4M
>
>
> Deployment implications. SAGE-based simulation and multi-turn trajectory rollouts are used only during training. At deployment time, the RLVER-enhanced model performs standard single-pass inference: it takes a dialogue context as input and produces one response, without inner thoughts, auxiliary rollouts, or simulator calls. Thus, inference latency and per-request cost remain essentially identical to the base LLM, and the method scales cleanly to real-world deployment.

---

> > ### Comment · Reviewer_NdQV · 2025-11-25
> >
> > Thank you for the very thorough and well-organized response, and for running a substantial number of additional experiments.
> >
> > Several of my main concerns are now convincingly addressed. The new evaluations beyond SAGE—covering open-domain chit-chat, ESConv with both automatic metrics and human judgments, and the added human study along the five empathy-related dimensions—substantially strengthen the case for external validity and reduce my worries about self-referential evaluation. The correlation analysis between human ratings, SAGE scores, and LLM-as-a-judge metrics is also reassuring, suggesting that the reported gains are not merely artifacts of the simulator.
> >
> > I also appreciate the new results on Llama-3.1-8B-Instruct, which demonstrate that RLVER is not tied to a single base model or architecture, and the SYCON BENCH evaluation together with the outcome-level reward design, which provide concrete evidence that the method does not simply increase sycophancy. Finally, the explicit accounting of token/sample costs and the clarification that training-time simulation does not affect deployment-time latency help make the scalability story much clearer.
> >
> > That said, the rebuttal substantially improves my confidence in the robustness, generality, and practical relevance of the approach. I am therefore raising my overall score.

---

> > > ### Author Response · Authors · 2025-11-26
> > >
> > > Thank you for your positive follow-up, for taking the time to review our response, and for updating your evaluation. We sincerely appreciate your thoughtful comments, particularly regarding safety risks and external validity. Your feedback directly motivated the inclusion of new validation experiments and robustness checks, which have significantly strengthened the paper.  We fully commit to incorporating all the discussed revisions—including the new experiments, clarifications, and detailed analyses—into the final version of the manuscript to ensure it is comprehensive and rigorous.

---

### Official Review · Reviewer_TNY1 · 2025-10-30

**Soundness:** 2
**Presentation:** 2
**Contribution:** 3
**Rating:** 4
**Confidence:** 4

**Summary:**

This paper proposes RLVER, a reinforcement learning framework designed to enhance the emotional intelligence (EQ) of large language models (LLMs) by optimizing them with verifiable emotion rewards. The framework leverages the SAGE (Sentient Agent as a Judge) simulator — an LLM-driven environment that models user affect and generates deterministic emotion scores — to provide reward signals for PPO and GRPO training. The authors report substantial improvement on the Sentient Benchmark (13.3 → 79.2) while maintaining performance on math, coding, and instruction-following tasks.
The idea of integrating psychologically-grounded verifiable feedback into RL for emotion modeling is promising. However, the current paper lacks methodological clarity, validation rigor, and experimental diversity to support its central claims.

**Strengths:**

- The paper explores a novel and meaningful problem — improving emotional intelligence in LLMs via reinforcement learning.

- The notion of verifiable emotion rewards represents a new attempt to make affective reinforcement signals interpretable and consistent.

- The reported improvements on the Sentient Benchmark are substantial and suggest potential in empathy-oriented RL training.

**Weaknesses:**

- The method section, particularly Figure 1, is hard to interpret. The mapping between variables and their roles in the RL process is unclear. No pseudocode or formal algorithmic steps are provided, making it difficult to reproduce or verify the approach.
- No validation is presented comparing SAGE-generated emotion scores with human judgments or psychological gold standards. This undermines the “verifiable” aspect of the proposed reward, as it lacks empirical reliability testing.
- Experiments are conducted only on Qwen2.5-7B-Instruct, a single Chinese-centric open model. The paper lacks evidence that RLVER generalizes to different architectures or scales (e.g., LLaMA, Mistral, Gemma). This severely limits the generalizability and reproducibility of the method.
- No comparison is made with other empathy or affective dialogue datasets, such as EmpatheticDialogues, ESConv, or MELD. As a result, it is unclear whether RLVER’s improvements translate beyond its own simulation environment.


- All results are derived from simulated environments, with no human-in-the-loop testing. Without qualitative or quantitative human evaluation, the claim that RLVER produces more empathetic and emotionally aligned behavior remains unverified.

**Questions:**

- Provide detailed definitions for all mathematical symbols.
-  Add an algorithm box or pseudocode that explicitly connects the emotion score computation with PPO/GRPO updates.
- Include human evaluation comparing SAGE emotion scores with real human annotations. Report quantitative metrics (e.g., correlation, agreement rates) to establish reward fidelity.
- Explore different reward designs (per-turn, cumulative, delta-based)

---

> ### Author Response · Authors · 2025-11-24
> **Response to Reviewer TNY1 (Part 1/5)**
>
> We sincerely appreciate your recognition of the conceptual novelty and the promising nature of verifiable emotion rewards in RL. We have addressed your concerns regarding clarity, validation, and experimental diversity with extensive new experiments and revisions below.
>
>
> ---
>
> ### W1&Q1&Q2: Notation clarity and Method Reproducibility
> **1. Expanded Figure Caption & Notation & Algorithm:** We agree that clarity in the method section is important. In the revised manuscript, we have expanded Figure 1's caption and text to explicitly map all variables to their roles. We have also added a dedicated Algorithm Box in Appendix F, providing pseudocode that formally defines the transformation from SAGE emotion scores to PPO/GRPO updates.
>
> **2. Reproducibility:** We reaffirm that our complete training code, simulator interactions, and reward computation logic are available in the [anonymous repository](https://anonymous.4open.science/r/RLVER-49F6) linked in the paper.
>
>
>
>
> ---
>
>
>
> ### W2&Q3: Validation and empirical reliability testing of SAGE
>
> We thank the reviewer for raising this point. We address the concern about "no validation" with three levels of evidence (including two new experiments added to Appendix E):
>
> #### 1. Validation of SAGE's Determinism (Existing)
> SAGE's reliability is well-supported by psychological [1] and human validation. The emotion scores exhibit a strong Pearson correlation ($r=0.82$) with the Barrett–Lennard Relationship Inventory (BLRI) and correlate highly with utterance-level empathy metrics ($r \approx 0.79$). Furthermore, SAGE achieves high consistency with **human simulations** (85.3%) and a high reasonableness rating (89.5%) (Table 2 in SAGE), confirming it aligns closely with **human-validated** affective constructs rather than being arbitrary.
>
>
> #### 2. Further human evaluation (New)
> To further address concerns regarding reliance on SAGE, we conducted an independent human evaluation on the generated emotional-change predictions (details added to Appendix E.1). Ten annotators each evaluated 20 dialogue contexts. The results show that SAGE achieves **82.65%** consistency with human judgments and a **88.35%** reasonableness rating, demonstrating the effectiveness and validity of SAGE.
>
> #### 3. Stability Assessment (New)
> Empirically, SAGE’s predictions are stable under our settings. To directly stress-test determinism, we conducted an additional experiment (added to Appendix E.3). We randomly selected 100 dialogue contexts from the test set and had the agent infer the emotional change 10 times for each context. The direction of the emotional change was consistent **90.2%** of the time, directly confirming the robustness of the persona-driven reasoning.
>
> These results collectively demonstrate that the "verifiable" reward signal is empirically reliable and grounded in human affect.
>
> [1] Zhang, Bang, et al. "Sentient Agent as a Judge: Evaluating Higher-Order Social Cognition in Large Language Models." arXiv preprint arXiv:2505.02847 (2025).
>
>
>
> ---
>
> ### W3: Generalization to Different Architectures
>
> To address the concern about reliance on Qwen2.5 (Chinese-centric), we conducted a new experiment applying RLVER (PPO-Thinking & PPO-Non-Thinking) to Llama-3.1-8B-Instruct, a widely used English-centric architecture. We kept the experimental pipeline identical to the original setup.
>
> 1. Performance on Sentient Benchmark & Chit-Chat
>
> | Model               | Sentient Benchmark (Score) | Sentient Benchmark (Success) | Sentient Benchmark (Failure) | Chit Chat (Score) | Chit Chat (Success) |  Chit Chat (Failure)     |
> |---------------------|----------------|--------------|--------------------|--------------------|-------------------|-------------------|
> | Llama-3.1-8B-Instruct | 37.6          | 4%       | 43%              | 44.7               | 29%              | 49%       |
> | PPO-thinking        | 65.8         | 18%        | 13%              | 78.7              | 56%             |   9%    |
> | PPO-non-thinking        | 49.1         | 20%        | 38%              | 47.9              | 31%             |   46%    |
>
> 2. Core Capabilities evaluated by DeepSeek-R1-0528 (averaged of 3 runs)
>
> | Model               | Empathic Depth | Core Insight | Solution Crafting | Dialogue Guidance | Style Flexibility |
> |---------------------|----------------|--------------|--------------------|--------------------|-------------------|
> | Llama-3.1-8B-Instruct | 2.983          | 3.047        | 3.120              | 3.290               | 2.963             |
> | PPO-thinking        | 3.517          | 3.522        | 3.892             | 3.940              | 3.670             |
> | PPO-non-thinking        | 3.125          | 3.283        | 3.418             | 3.204              | 2.919             |
>
> These results confirm that RLVER effectively generalizes to different model architectures, demonstrating that the framework is model-agnostic. The new experiment has been added to Appendix I in the revised manuscript.

---

> ### Author Response · Authors · 2025-11-24
> **Response to Reviewer TNY1 (Part 2/5)**
>
> ### W4: RLVER’s improvements beyond its own simulation environment.
> We agree with you that it is important to validate RLVER's generalization beyond the trained simulation environment. We address the concern in three ways:
>
> #### 1. Evaluation on open-domain Chit-Chat:
> As shown in Table 1, RLVER yields consistent improvements in an out-of-domain, unconstrained chit-chat, proving that it learns general interaction skills beyond the emotional support environment.
>
>
> #### 2. Generalization across different base models
> We further evaluated our models using Gemini-2.5, Gemini-Think, and GPT-4o as independent base reasoning models as the simulator. The PPO-Think model consistently outperforms baselines across all external simulators, which proves the robustness of our results to single judge bias.
>
> |  | Gemini2.5 | Gemini2.5-Think |GPT4o|
> | -------- | -------- | -------- |-------|
> | Qwen2.5-7B-Instruct     | 11.96     |  13.77   |12.51   |
> | PPO-unthink   |  73.03   | 63.08     |65.63|
> | PPO-think     | 84.30    | 90.60    |82.29|

---

> ### Author Response · Authors · 2025-11-24
> **Response to Reviewer TNY1 (Part 3/5)**
>
> ### W4: RLVER’s improvements beyond its own simulation environment.
> #### 3. Evaluation on External Empathy Dataset
> We appreciate the suggestion to benchmark against established datasets like EmpatheticDialogues, ESConv, and MELD. During our research, we carefully examined these datasets but found that most existing empathy or affective datasets focus on single-turn evaluation or empathy understanding (action choosing) in multiple-choice format [1, 2, 4, 5], or rely on surface-level textual metrics such as BLEU or perplexity[3, 6], which neither align with our multi-turn dialogue setting nor reflect real-world interaction scenarios.
>
> **Evaluation on ESConv dataset (Automatic Metrics)**
> ESConv [3] is widely recognized for multi-turn emotional support. However, its evaluation protocol relies mainly on lexical metrics (PPL, BLEU, ROUGE, Extrema), which are insufficient for assessing emotional-support quality in today's open-ended dialogue evaluation.
> To assess generalization, we evaluated our models under the official ESConv setup:
>
> | Model              | PPL | BLEU-2  | ROUGE-L |  Extrema |
> |--------------------|--------|--------|---------|---------|
> | **Qwen2.5-7B-Instruct** | 6.39 | 3.61   |  12.71   | 89.68   |
> | **PPO-Non-Thinking**    | 27.84 | 3.98   |  13.31     | 90.03   |
> | **PPO-Thinking**        | 10.67 | 4.16   | 11.77    | 88.11   |
>
> We observe a clear mismatch between lexical overlap metrics (BLEU, ROUGE, PPL) and semantic similarity metrics (Extrema). Although lexical similarity is low, all models exhibit high semantic alignment with reference responses—consistent with the nature of emotional-support conversations where diverse paraphrases are equally valid. This demonstrates that existing lexical metrics cannot faithfully reflect model quality in our task, and further motivates the use of LLM-as-a-Judge and human evaluation.
>
>
> **Human evaluation under ESConv dimensions**
> Given the limitations of n-gram metrics, we further conducted a Human Evaluation following the ESConv paper. We recruited 10 paid annotators to each evaluate 10 cases based on the four official ESConv dimensions (Fluency, Identification, Comforting, Suggestion). Win–lose counts (ties omitted) are:
>
> | Qwen2.5-7B-Instruct    vs. | PPO-Non-Thinking Win | PPO-Non-Thinking Lose | PPO-Thinking Win | PPO-Thinking Lose |
> |-----------|------------|-------------|--------------|---------------|
> | Fluency       | 55 | 7 |59  |12  |
> | Identification| 70 |19 | 83 |7 |
> | Comforting    |71 |17  | 78 |7 |
> | Suggestion    | 79 | 10 | 71 |20  |
> | Overall       | 92 | 3 | 90 |2  |
>
> (Note: "Win" indicates the PPO model was judged superior to the Qwen2.5-Base model)
>
> **Conclusion:** Human judgments conclusively show that RLVER-trained models outperform the base model across all ESConv dimensions. This confirms that the empathetic capabilities learned via RLVER are not overfitted to SAGE but successfully generalize to established external benchmarks and real-world conversation standards.
>
>
> Together, these results demonstrate that RLVER is not tied to the SAGE environment and that its improvements translate beyond its own simulation environment.
>
> [1] Sabour, Sahand, et al. "Emobench: Evaluating the emotional intelligence of large language models." Proceedings of the 62nd Annual Meeting of the Association for Computational Linguistics (Volume 1: Long Papers). 2024.
>
> [2] Chen, Zhuang, et al. "Tombench: Benchmarking theory of mind in large language models." arXiv preprint arXiv:2402.15052 (2024).
>
> [3] Liu, Siyang, et al. "Towards emotional support dialog systems." arXiv preprint arXiv:2106.01144 (2021).
>
> [4] Zahiri, Sayyed M., and Jinho D. Choi. "Emotion Detection on TV Show Transcripts with Sequence-Based Convolutional Neural Networks." AAAI Workshops. Vol. 18. 2018.
>
> [5] Paech, Samuel J. "Eq-bench: An emotional intelligence benchmark for large language models." arXiv preprint arXiv:2312.06281 (2023).
>
> [6] Rashkin, Hannah, et al. "Towards empathetic open-domain conversation models: A new benchmark and dataset." Proceedings of the 57th annual meeting of the association for computational linguistics. 2019.

---

> ### Author Response · Authors · 2025-11-24
> **Response to Reviewer TNY1 (Part 4/5)**
>
> ### W5&Q3: Human-in-the-loop testing
> We agree that human validation is essential. In the revised version, we conducted a new Human-in-the-Loop Evaluation (added to Appendix E.2) to directly validate the effectiveness of our framework.
>
> **Protocol:** We recruited 15 paid adult annotators, each evaluate 20 dialogue cases across five empathy-related dimensions. Following the ESConv [1] protocol, we employed blind pairwise comparisons: annotators compared the outputs of the three models (Qwen2.5-7B-Instruct (base), PPO-non-thinking, PPO-thinking) for the same context and selected the superior response for each dimension.
>
> **Results:** As shown in the tables below, RLVER-trained models achieved significantly higher human win rates compared to the base model. Crucially, these human win rates exhibit a very strong correlation with our automated LLM-as-a-Judge results (Spearman $\rho = 0.899$, $p < 10^{-5}$), confirming that our reported improvements are robust and perceptually valid to humans.
>
> ##### LLM Evaluation Scores
> | Model               | Empathic Depth | Core Insight | Solution Crafting | Dialogue Guidance | Style Flexibility |
> |---------------------|----------------|--------------|--------------------|--------------------|-------------------|
> | Qwen2.5-7B-Instruct | 2.239          | 2.388        | 2.762              | 2.494              | 1.885             |
> | PPO-thinking        | 3.971          | 3.623        | 3.342              | 3.677              | 3.285             |
> | PPO-non-thinking    | 3.247          | 3.099        | 3.830              | 3.550              | 3.098             |
>
> ##### *Win Rate by Dimension*
> | Model               | Empathic Depth | Core Insight | Solution Crafting | Dialogue Guidance | Style Flexibility |
> |---------------------|----------------|--------------|--------------------|--------------------|-------------------|
> | Qwen2.5-7B-Instruct | 10.0%          | 13.3%        | 10.7%              | 8.7%               | 9.3%              |
> | PPO-thinking        | 69.3%          | 67.3%        | 49.3%              | 66.0%              | 63.3%             |
> | PPO-non-thinking    | 62.0%          | 58.0%        | 76.7%              | 58.7%              | 58.7%             |
>
> ##### *Correlation with LLM-as-Judge*
> | Metric        | Value     |
> |---------------|-----------|
> | Spearman ρ    | 0.899     |
> | p-value       | 0.000005  |
>
>
> This confirms that the improvements brought by RLVER go well beyond simulation effects and are robustly supported by human evaluators. Additional details are provided in Appendix E.2.
>
> [1]Liu, Siyang, et al. "Towards emotional support dialog systems." arXiv preprint arXiv:2106.01144 (2021).

---

> ### Author Response · Authors · 2025-11-24
> **Response to Reviewer TNY1 (Part 5/5)**
>
> ### Q4: Ablation Study on Reward Designs
> We appreciate the suggestion to explore different reward structures. We agree that investigating reward granularity improves the completeness of our work. Therefore, we conducted ablation studies on the PPO-Thinking model with two alternative settings:
> 1. Setting 1: Turn-level Reward Only.
> In this setting, we assign a turn-level reward defined as $r_t = (e_t - e_{t-1}) / 10$. Here, the immediate emotional change serves as the reward. We empirically scale by 10 because observed emotional changes typically fall within the $[-10, 10]$ range, normalizing the reward to roughly $[-1, 1]$.
> 2. Setting 2: Turn+Outcome Reward.
> In this setting, we explore a trade-off by assigning a reward defined as the average score of of the turn-level and outcome-level reward: $r_{t\phi}=(r_t+r_\phi)/2$.
>
>
> |Model	|Sentient Score|	Chit-Chat|	Empathic Depth	|Core Insight|	Solution Crafting|	Dialogue Guidance|	Style Flexibility|
> |---------------------|----------------|--------------|--------------------|--------------------|-------------------|-------------------|-------------------|
> |Qwen2.5-7B-Instruct	|13.3	|37.8		|2.333	|2.537|	2.898|	2.594|1.898|
> |PPO-Think (Original / Outcome Only)|	79.2|	62.1		|3.727|	3.455|	3.187	|3.526|	3.033|
> |PPO-Think (Turn Only)	|74.7	|47.6|		2.543|	2.820|	3.378|	3.170	|2.373|
> |PPO-Think (Turn + Outcome)|	81.2|	59.8	|	3.422|	3.242|3.720	|3.683	|3.067|
>
> (The five core capabilities are evaluated by DeepSeek-R1-0528 (averaged of 3 runs))
>
> We found that:
> 1. **"Outcome Only" is superior for developing Deep Empathy and Core Insight**: Our original design achieves the highest scores in Empathic Depth (3.727) and Core Insight (3.455). This supports our hypothesis that high-level empathy is a long-horizon objective that requires holistic insight and empathic behavior. In contrast, optimizing purely for immediate turn-level reactions ("Turn Only") may lead to myopic behaviors, significantly degrading deep insight and empathy (dropping to 2.820 and 2.543).
> 3. **"Turn + Outcome Trade-off" helps improve personalized support**: Incorporating turn-level rewards ("Turn + Outcome") slightly boosts the overall Sentient Score (81.2), significantly improves Solution Crafting (3.720) and slightly improves Dialogue Guidance (3.683). This suggests that the trade-off to add dense, immediate feedback helps the model formulate more concrete, actionable strategies for users, though at a slight cost to deep introspective insight.
>
> These results further illustrate the validity and flexibility of our RLVER framework. We have added this insightful comparison to Appendix H in the revised manuscript. Thanks again for your constructive feedback.

---

> > ### Comment · Reviewer_TNY1 · 2025-11-25
> >
> > Thank you for your responses; they have addressed some of my concerns. I have updated my score accordingly and hope it can be reviewed in the final revision.

---

> > > ### Author Response · Authors · 2025-11-26
> > >
> > > Thank you for your thoughtful follow-up, for taking the time to review our rebuttal, and for updating your evaluation. We sincerely appreciate your close reading of the methodology. Your constructive suggestions enabled us to clarify the theoretical framework, strengthen the validation experiments, and significantly improve the paper's rigor. We are grateful for the time and effort you dedicated to improving our work.

---

### Official Review · Reviewer_HwhH · 2025-11-01

**Soundness:** 2
**Presentation:** 3
**Contribution:** 2
**Rating:** 6
**Confidence:** 4

**Summary:**

The paper introduces RLVER (Reinforcement Learning with Verifiable Emotional Rewards) - a reinforcement learning framework designed to improve the emotional intelligence of large language models (LLMs) through verifiable reward signals generated by a simulated user named SAGE.
Using PPO and GRPO on top of Qwen2.5-7B-Instruct, the authors train the model to optimize for “emotional scores” in dialogue, measured via a new benchmark called Sentient Benchmark. The best configuration (PPO + “thinking” mode) achieves a jump from 13.3 to 79.2, approaching the level of frontier proprietary models while maintaining stable mathematical and coding capabilities.
The study further examines the effects of simulated user complexity, the impact of chain-of-thought (“thinking”) prompts, and includes a multi-competency evaluation using “LLM-as-a-Judge”.

**Strengths:**

- Originality: Clear conceptual innovation — reinforcement learning guided by verifiable emotional scores from a controllable simulator (SAGE).
- Quality: Demonstrates substantial improvement (Sentient 79.2 vs 13.3 baseline) while preserving general abilities in math and code.
- Clarity: Writing is structured; diagrams effectively communicate architecture and reward flow.
- Significance: Addresses an underexplored but crucial domain — emotional intelligence in LLMs - with a scalable, privacy-safe RL training loop.
- Analytical depth: Includes ablations across RL methods (PPO vs GRPO), analysis of “thinking” format, and breakdown by emotional competencies.

**Weaknesses:**

- Inconsistent reward definition. Emotional scores are defined in [0, 100], yet Table 1 states success as “score > 100” - logically impossible. Likely an error needing correction.
- Unclear determinism of SAGE simulator. The simulator is described as deterministic but implemented via a generative model (DeepSeek-V3). The temperature and decoding strategy for reproducible rewards are unspecified.
Overreliance on one simulator. All benchmarks (Sentient, chit-chat) derive from SAGE, raising concerns of overfitting and limited external validity without human evaluation.
- LLM-as-a-Judge limitations. Despite reported inter-rater consistency, subjective competence scoring by other LLMs risks circularity and bias. Human validation is missing.
- Ambiguity in setup details. Dialogue length differs between text (10 turns) and appendix (8 turns). Hyperparameters and compute cost are under-reported.
- Lack of robustness tests. No ablations with alternate simulators or format-reward removal, which would confirm generality.

**Questions:**

1. Please clarify the reward success threshold (Table 1) and ensure consistency with the [0–100] range.
2. Specify the exact number of dialogue turns used for training and evaluation (8 or 10?).
3. How is determinism achieved in SAGE reward generation (temperature? seed?)? Provide variance statistics if re-evaluated multiple times.
4. Did you perform any human evaluation (blind pairwise or Likert rating) to validate emotional quality outside SAGE?
5. Can you report robustness to simulator replacement (e.g., using a different model for rewards)?

---

> ### Author Response · Authors · 2025-11-24
> **Response to Reviewer HwhH (Part 1/3)**
>
> We sincerely appreciate your recognition of our work's significance, particularly regarding the conceptual novelty, strong empirical improvements, and comprehensive analysis. We are also grateful for your constructive feedback on the presentation details. We address your specific questions point-by-point below.
>
>
> ---
>
> ### W1&Q1: Clarification on Reward Success Threshold
>
> We apologize for the confusion caused by the typographical error in Table 1. As noted in the paper, the emotion score is strictly normalized to the range $[0, 100]$. While the simulator predicts internal emotional changes that might exceed this bound (e.g., previous score 95 with a +10 change), the final reward is clipped at the upper bound 100. Therefore, "Success" is defined as achieving the maximum possible score (i.e., score $= 100$). The notation "$> 100$" in the table caption was a typo and we will correct this in the final revision. Thank you for pointing out.
>
> ---
> ### W2&Q3&Q4: Determinism of SAGE simulator
> We appreciate your concern regarding determinism. As described in the paper, each emotional update produced by SAGE is computed through a transparent multi-step reasoning process grounded in the simulator’s persona, conversational history, contextual factors, and scenario-specific goals.
>
> #### 1. Validation of SAGE's Determinism (Existing)
> SAGE's reliability is well-supported by psychological[1] and human validation. The emotion scores exhibit a strong Pearson correlation ($r=0.82$) with the Barrett–Lennard Relationship Inventory (BLRI) and correlate highly with utterance-level empathy metrics ($r \approx 0.79$). Furthermore, SAGE achieves high consistency with human simulations (85.3%) and a high reasonableness rating (89.5%) (Table 2 in SAGE), confirming it aligns closely with human-validated affective constructs rather than being arbitrary.
>
>
> #### 2. Further human evaluation (New)
> To further address concerns regarding reliance on SAGE, we conducted an independent human evaluation on the generated emotional-change predictions (details added to Appendix E.1). Ten annotators each evaluated 20 dialogue contexts. The results show that SAGE achieves 82.65% consistency with human judgments and a 88.35% reasonableness rating, demonstrating the effectiveness and validity of SAGE.
>
> #### 3. Stability Assessment
> Empirically, SAGE’s predictions are stable under our settings. To directly stress-test determinism, we conducted an additional experiment (added to Appendix E.3). We randomly selected 100 dialogue contexts from the test set and had the agent infer the emotional change 10 times for each context. The direction of the emotional change was consistent 90.2% of the time, directly confirming the robustness of the persona-driven reasoning.
>
> #### 4. Cross-validation with Base Reasoning Models
> To address the concern that benchmarks rely solely on SAGE, we further evaluated our trained models using independent simulators not used during training: Gemini 2.5, Gemini 2.5-Think, and GPT-4o. Each model serves as an alternative emotional-reasoning base model.
>
> |  | Gemini2.5 | Gemini2.5-Think |GPT4o|
> | -------- | -------- | -------- |-------|
> | Qwen2.5-7B-Instruct     | 11.96     |  13.77   |12.51   |
> | PPO-non-thinking   |  73.03   | 63.08     |65.63|
> | PPO-thinking     | 84.30    | 90.60    |82.29|
>
> Across these external simulators, our model consistently outperforms the baselines. This demonstrates that the learned behaviors generalize beyond SAGE and are not tied to a single simulator’s inductive biases.
>
> #### 5. Reproducibility & Hyperparameter Choice
> In our implementation, we query the DeepSeek-V3 API with temperature=0.5 (with top-p=1.0). This setting was chosen to strike a critical balance between reasoning stability and interaction diversity. While it ensures the direction of emotional change remains highly consistent (as evidenced by the 90.2% stability reported in Point 3 above), it intentionally retains a degree of stochasticity to mimic the natural variation in human feedback and actions. This prevents the RL agent from overfitting to deterministic string patterns, ensuring it learns robust strategies applicable to diverse human expressions. All settings are fully specified in our [anonymous repository](https://anonymous.4open.science/r/RLVER-49F6) linked in the paper.
>
> [1] Zhang, Bang, et al. "Sentient Agent as a Judge: Evaluating Higher-Order Social Cognition in Large Language Models." arXiv preprint arXiv:2505.02847 (2025).

---

> > ### Comment · Reviewer_HwhH · 2025-11-25
> >
> > Thanks for the clarifications — they were helpful and addressed my concerns. I’ll retain my score, and I’m still convinced that my relatively high evaluation accurately reflects the quality of the work.

---

> > > ### Author Response · Authors · 2025-11-26
> > >
> > > Thank you for your constructive feedback and for taking the time to review our rebuttal. We appreciate your recognition of the core ideas and empirical strengths of our work. Your comments were instrumental in helping us refine the clarity, strengthen the robustness analyses, and improve the overall presentation. We are grateful for your effort in guiding us to improve this paper.

---

> ### Author Response · Authors · 2025-11-24
> **Response to Reviewer HwhH (Part 2/3)**
>
> ### W3&Q4: Human validation
> While LLM-as-a-Judge is becoming a standard practice in recent research, we agree that human evaluation remains the gold standard for subjective dialogue tasks. As noted in Response to W2, SAGE aligns well with human judgment, providing a stable and interpretable automated evaluation signal. To more directly validate our model's performance, we further conducted a new Human-in-the-Loop Evaluation (added to Appendix E.2).
>
> **Protocol:** We recruited 15 paid adult annotators, each evaluate 20 dialogue cases across five empathy-related dimensions. Following the ESConv [1] protocol, we employed blind pairwise comparisons: annotators compared the outputs of the three models (Qwen2.5-7B-Instruct (base), PPO-non-thinking, PPO-thinking) for the same context and selected the superior response for each dimension.
>
> **Results:** As shown in the tables below, RLVER-trained models achieved significantly higher human win rates compared to the base model. Crucially, these human win rates exhibit a very strong correlation with our automated LLM-as-a-Judge results (Spearman $\rho = 0.899$, $p < 10^{-5}$), confirming that our reported improvements are robust and perceptually valid to humans.
>
> ##### LLM Evaluation Scores
> | Model               | Empathic Depth | Core Insight | Solution Crafting | Dialogue Guidance | Style Flexibility |
> |---------------------|----------------|--------------|--------------------|--------------------|-------------------|
> | Qwen2.5-7B-Instruct | 2.239          | 2.388        | 2.762              | 2.494              | 1.885             |
> | PPO-thinking        | 3.971          | 3.623        | 3.342              | 3.677              | 3.285             |
> | PPO-non-thinking    | 3.247          | 3.099        | 3.830              | 3.550              | 3.098             |
>
> ##### *Win Rate by Dimension*
> | Model               | Empathic Depth | Core Insight | Solution Crafting | Dialogue Guidance | Style Flexibility |
> |---------------------|----------------|--------------|--------------------|--------------------|-------------------|
> | Qwen2.5-7B-Instruct | 10.0%          | 13.3%        | 10.7%              | 8.7%               | 9.3%              |
> | PPO-thinking        | 69.3%          | 67.3%        | 49.3%              | 66.0%              | 63.3%             |
> | PPO-non-thinking    | 62.0%          | 58.0%        | 76.7%              | 58.7%              | 58.7%             |
>
> ##### *Correlation with LLM-as-Judge*
> | Metric        | Value     |
> |---------------|-----------|
> | Spearman ρ    | 0.899     |
> | p-value       | 0.000005  |
>
>
> This confirms that the improvements brought by RLVER go well beyond simulation effects and are robustly supported by human evaluators. Additional details are provided in Appendix E.2.
>
>
> [1] Liu, Siyang, et al. "Towards emotional support dialog systems." arXiv preprint arXiv:2106.01144 (2021).
>
> ---
>
> ### W4&Q2: Disambiguation and cost report
>
> #### 1. Clarification on Dialogue Turns
> Thank you for pointing out the ambiguity. The mention of "10 turns" in §3.1  was a clerical error from an earlier draft. In our actual implementation and all reported experiments, the maximum dialogue length was fixed at 8 turns, as correctly stated in Appendix A.3. We will correct the main text to consistently state "8 turns" to avoid confusion.
>
> #### 2. Hyperparameters and Compute Cost
> All training-related hyperparameters are fully provided in Appendix A.3 (learning rate, batch size, warm-up, rollout count, template choice, simulator model, temperature settings, etc.).
>
> For training cost, we take PPO training as an example, each on-policy update samples M = 32 dialogue trajectories. Each trajectory consists of 8 turns, and on average consumes:
>
> User (Simulator) API:
>
> average input tokens: 32,496.41
>
> average output tokens: 2,559.04
>
> Assistant（Qwen2.5-7B-Instruct）:
>
> average input tokens: 2,390.62
>
> average output tokens: 383.55
>
> PPO performs 350 RL updates, yielding 32 × 350 = 11,200 trajectories in total.
> The token usage is therefore:
>
> Simulator input tokens:
> 11,200 × 32,496.41 ≈ 364M
>
> Simulator output tokens:
> 11,200 × 2,559.04 ≈ 29M
>
> Assistant input tokens:
> 11,200 × 2,390.62 ≈ 27M
>
> Assistant output tokens:
> 11,200 × 383.55 ≈ 4M

---

> ### Author Response · Authors · 2025-11-24
> **Response to Reviewer HwhH (Part 3/3)**
>
> ### W5&Q5: Robustness tests.
> Thank you for highlighting the importance of evaluating robustness under alternative simulator settings. We note that the original paper already includes results demonstrating robustness to environment and reward variations, such as the comparison between vanilla and challenging user simulators. To futher address this concern, we additionally trained our simulator using GPT-4o as the base model, without changing any other components of our pipeline. The downstream models trained with GPT-4o rewards exhibit similar performance gains, comparable to our main results:
> |  | Sentient Benchmark|
> | -------- | -------- |
> | Qwen2.5-7B-Instruct     | 11.96     |
> | PPO-unthink   |  64.40  |
> | PPO-think     | 61.15   |
>
> These findings provide strong evidence that the proposed approach is robust to simulator choice. The emotional-dynamics reward signal learned by our framework generalizes effectively and is not dependent on a specific simulator's inductive bias.

---

### Author Response · Authors · 2025-11-24
**General Response to Reviewers**

We sincerely thank all reviewers for their constructive feedback and for recognizing the conceptual novelty, empirical strength, and broader impact of our work. In response, we have strengthened the paper along six dimensions:

1. **Conceptual clarity:** We expanded Figure 1 and revised its caption to explicitly map all variables to their RL roles. Additionally, we added a dedicated Algorithm Box (Appendix F) to formally describe the proposed method and corrected minor inconsistencies in hyperparameter reporting.

2. **Validation of the Simulator(Appendix E):** We reinforced the reliability of SAGE by adding:
(1) human validation of emotional-change predictions;
(2) a stability test across repeated queries.
We also consolidated evidence demonstrating SAGE’s consistency with human judgments and its psychological grounding.


3. **External validation:** We expanded our evaluation to include:
(1) Human evaluation of RLVER-trained models (Appendix E.2);
(2) Evaluation on the ESConv dataset (both automatic metrics and human judgments);
(3) Cross-simulator evaluation using Gemini-2.5, Gemini-Think, and GPT-4o.


4. **Robustness of the RL framework:** We conducted new experiments to verify robustness, including:
(1) an ablation study on reward designs (outcome-only, turn-only, hybrid) in Appendix H;
(2) a cross-architecture study on Llama-3.1-8B-Instruct in Appendix I.

5. **Sycophancy-related Analysis (Appendix G):** We addressed sycophancy-related concerns by:
(1) Clarifying how the hidden intention and outcome-level rewarding mechanisms prevent blind agreement;
(2) adding an evaluation on SYCON BENCH [2], confirming that stance consistency is preserved.

6. **Cost reporting:** We provided a complete token-level accounting of rollout costs to ensure transparency and reproducibility.

Overall, these revisions provide a more comprehensive picture of RLVER’s robustness, generalization, safety, and efficiency, and additional external-validity evidence and human evaluations to further enrich the overall evaluation. We are grateful for the reviewers' insightful feedback, which helps significantly improve the quality of our work.

[1] Liu, Siyang, et al. "Towards emotional support dialog systems." arXiv preprint arXiv:2106.01144 (2021).

[2] Hong, Jiseung, et al. "Measuring Sycophancy of Language Models in Multi-turn Dialogues." arXiv preprint arXiv:2505.23840 (2025).

---

### Meta-Review · Area_Chair_TAUr · 2025-12-22

**Summary:**

The paper describes an approach to improve the emotional intelligence of LLMs by reinforcement learning from verifiable emotion rewards.  The reviewers expressed the following concerns:

1. Inconsistent reward definition
2. Unclear determinism of SAGE simulator
3. Human validation is missing
4. Ambiguity in setup details
5. Lack of robustness tests
6. The method section is hard to interpret
7. The paper lacks evidence that RLVER generalizes to different architectures or scales (e.g., LLaMA, Mistral, Gemma).
8. No comparison is made with other empathy or affective dialogue datasets, such as EmpatheticDialogues, ESConv, or MELD
9. Limited methodological novelty: No new RL objective/optimizer; primarily an engineering/system contribution.
10. Safety & multi-objective trade-offs under-specified: Optimizing “feeling better” risks sycophancy or avoidance when boundary-setting or factual correction is needed.
11. Cost/scalability underreported: Multi-turn on-policy RL is expensive

**Reviewer Concerns:**

The authors submitted a thorough rebuttal with several new experiments that were added to the paper.  Overall, they addressed the reviewers' concerns.

**Reviewer Scores:**

Reviewer HwhH is satisfied by the rebuttal and the revised paper.  The reviewer indicated that he/she would maintain its positive score and its recommendation for acceptance.

Reviewers TNY1 and  NdQV are also satisfied by the rebuttal and the revised paper.  They both indicated that they would increase their scores.

Overall, this is nice work that advances the state of the art in LLM emotional intelligence.  The proposed approach is effective and the empirical results provide strong support.  This represents a valuable contribution to the field of LLMs.

---

### Decision · Program_Chairs · 2026-01-26

Accept (Poster)